# Selecting Influential Samples for Long Context Alignment via Homologous Models' Guidance and Contextual Awareness Measurement

## Abstract

The expansion of large language models to effectively handle instructions with extremely long contexts has yet to be fully investigated. The primary obstacle lies in constructing a high-quality long instruction-following dataset devised for long context alignment. Existing studies have attempted to scale up the available data volume by synthesizing long instruction-following samples. However, indiscriminately increasing the quantity of data without a well-defined strategy for ensuring data quality may introduce low-quality samples and restrict the final performance. To bridge this gap, we aim to address the unique challenge of long-context alignment, i.e., modeling the long-range dependencies for handling instructions and lengthy input contexts. We propose **GATEAU**, a novel framework designed to identify the influential and high-quality samples enriched with long-range dependency relations by utilizing crafted **Homologous Models' Guidance (HMG)** and **Contextual Awareness Measurement (CAM)**. Specifically, HMG attempts to measure the difficulty of generating corresponding responses due to the long-range dependencies, using the perplexity scores of the response from two homologous models with different context windows. Also, the role of CAM is to measure the difficulty of understanding the long input contexts due to long-range dependencies by evaluating whether the model's attention is focused on important segments. Built upon both proposed methods, we select the most challenging samples as the influential data to effectively frame the long-range dependencies, thereby achieving better performance of LLMs. Comprehensive experiments indicate that GATEAU effectively identifies samples enriched with long-range dependency relations and the model trained on these selected samples exhibits better instruction-following and long-context understanding capabilities.

## 1 Introduction

Large Language Models (LLMs) with large context windows (Du et al., 2022; Li et al., 2023; Chen et al., 2024b) have demonstrated impressive capabilities across a wide range of real-world tasks that involve extremely long contexts, such as long-document summarization and multi-document question answering (Bai et al., 2023). Recent works to build long-context LLMs mainly focus on broadening the context window via position encoding extension and continual pre-training on long text (Chen et al., 2023b; Pal et al., 2023; Peng et al., 2024; Xiong et al., 2024; Han et al., 2024).

Despite these advancements, few studies consider the alignment of long-context LLMs to leverage their capabilities in understanding long input contexts and following complex instructions. A primary obstacle lies in the difficulty of constructing a high-quality long instruction-following dataset for supervised fine-tuning (SFT). Annotating long instruction-following data tends to be much more challenging than short ones. Because it is non-trivial for annotators to understand an excessively long context and provide high-quality responses. For example, annotators might be tasked with writing a summary for a document containing more than 64k words based on the given instruction. To bypass this, Li et al. (2023); Tworkowski et al. (2023); Xiong et al. (2024) construct the long instruction-following dataset by concatenating short instruction-following samples. Nonetheless, simply concatenating unrelated samples may not effectively simulate the long-range dependencies required for long-context tasks. For long-context tasks, modeling long-range dependencies is crucial,

as such strong semantic dependencies benefit LLMs to understand long input contexts and generate high-quality responses (Chen et al., 2024a; Wu et al., 2024). To preserve the inherent long-range dependency relations in the collected samples, Yang (2023); Chen et al. (2024b); Bai et al. (2024) focus on synthesizing long instruction-following data. For instance, Bai et al. (2024) synthesizes 10k samples by employing Claude 2.1 (Anthropic., 2023), which supports a context window of 200k tokens, to get responses for the collected long documents. However, LLMs trained on these synthetic samples, even with sufficiently long contexts, may still struggle to model the long-range dependencies. This is because indiscriminately increasing the quantity of data without a well-defined strategy for ensuring data quality may introduce low-quality samples that lack long-range dependency relations, e.g., such samples may rely solely on the limited tokens preceding the instruction or may not need to use long input contexts to generate a correct response. Therefore, a critical question arises: ***How can we effectively select influential samples from a vast amount of synthetic long instruction-following data for long context alignment?***

Unfortunately, previous studies for selecting high-quality instruction-following data primarily concentrate on short samples (Li et al., 2024b; Xia et al., 2024). Consequently, these studies may not be effective for long context alignment, as they ignore the unique challenge in long context alignment, i.e., how to select the samples enriched with meaningful long-range dependency relations. As such, we introduce **GATEAU**, which consists of **Homologous Models' GuidAnce (HMG)** and **ConTExtual Awareness MeasUrement (CAM)**, to identify the influential long instruction-following samples enriched with long-range dependency relations to achieve better long context alignment. The two proposed methods aim to separately measure the difficulty of generating corresponding responses and understanding long input contexts due to the long-range dependencies.

Specifically, HMG measures the difficulty of generating corresponding responses due to the long-range dependencies, by comparing the perplexity scores of the response between two homologous models with different context windows (e.g., the perplexity scores from LLaMA-2-7B-base-4k (Together.ai, 2023) and LLaMA-2-7B-base-64k (Bai et al., 2024)). The idea behind HMG is that the primary difference between homologous models with varying context windows lies in their different capabilities for modeling long-range dependencies. Thus, the disparity in the perplexity scores can be interpreted as reflecting the difficulty of generating the response caused by the long-range dependencies. The larger disparity between the scores indicates more difficulties for LLM in response generation due to the long-range dependencies. We also introduce CAM to measure the difficulty of understanding the long input contexts due to long-range dependencies, as it is hard for LLMs to utilize crucial information hidden in extremely long contexts. We first calculate the importance score of different input segments concerning the given response and subsequently measure whether LLMs can pay more attention to more important segments. Should LLM's attention focus more on less important segments, it implies that it is hard for the LLM to comprehend the long input contexts correctly. Ultimately, we take the weighted sum of both results from two methods as the final criterion for ranking the data, selecting the most challenging samples as influential ones. When trained on these selected samples characterized by complex long-range dependency relations, LLMs could effectively model the long-range dependencies and achieve better instruction-following performance.

We conduct extensive experiments to evaluate the effectiveness of GATEAU, including long-context understanding benchmark (LongBench (Bai et al., 2023)), long instruction-following benchmark (LongBench-Chat (Bai et al., 2024)), short instruction-following benchmark (MT-Bench (Zheng et al., 2023)), and Needle in A HayStack test (Gkamradt, 2023). With the proposed GATEAU, significant performance boosts are observed by using selected samples, e.g., the model trained on only 10% samples of the dataset achieves better performance than the model trained on the full dataset.

## 2 RELATED WORK

**Long Context Alignment.** Aligning the LLMs with instruction-following data can ensure they understand user instructions and give high-quality responses, which has been extensively studied in short context scenarios (Taori et al., 2023; Wang et al., 2023a;b). However, excessively long contexts present unique challenges for long context alignment. Li et al. (2023); Tworkowski et al. (2023); Xiong et al. (2024) construct the long instruction-following dataset by concatenating short instruction-following samples. Yet, simply concatenating unrelated sentences may not effectively simulate the long-range dependency relations for long-context tasks. Thus, Yang (2023); Chen et al. (2024b);

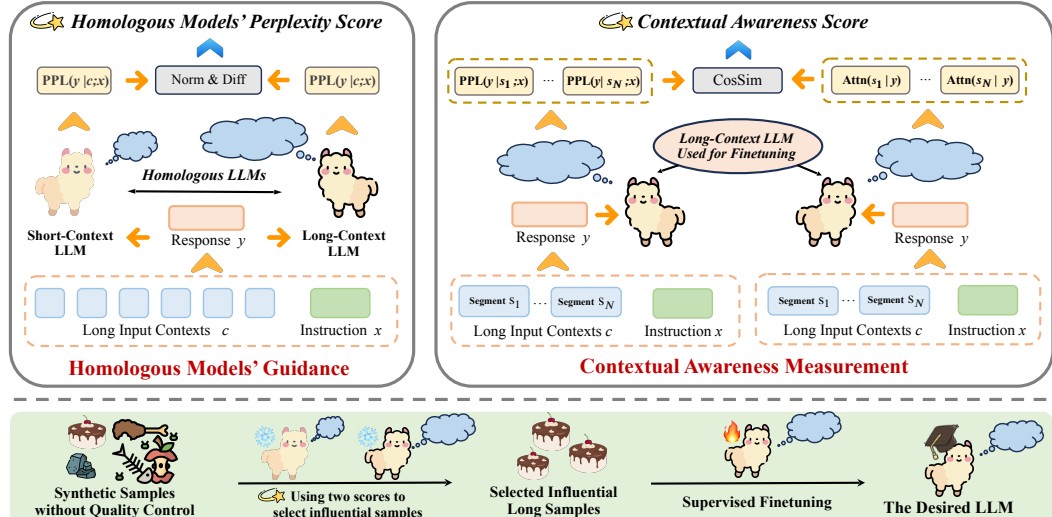

Figure 1: An overview of our framework **GATEAU**. Unlike directly training LLMs with the entire dataset, GATEAU first selects samples enriched with long-range dependency relations by using two proposed methods. Then it uses selected influential samples for training long-context LLMs.

Bai et al. (2024) construct long instruction-following data by collecting long-context materials as inputs and querying Claude to get the response. However, using these synthetic data without a clear strategy for ensuring data quality may lead to the inclusion of low-quality samples (e.g., samples without meaningful long-range dependency relations). Training LLMs on such low-quality samples can ultimately constrain their final performance.

**Data Selection for Alignment.** As Zhou et al. (2023) makes the statement that *less is more for alignment*, many works attempt to select influential and high-quality samples to empower the LLMs' instruction-following capabilities. Chen et al. (2023a); Liu et al. (2024) attempt to utilize the feedback from close-source LLMs (e.g., ChatGPT) to select samples. On the other hand, Cao et al. (2024); Li et al. (2024b); Ge et al. (2024); Xia et al. (2024) try to utilize the well-designed metrics (e.g., complexity) based on open-source LLMs to rank and select the samples. Meanwhile, Li et al. (2024c); Zhang et al. (2024) attempt to utilize the guidance from in-context learning. However, these methods only focus on selecting short instruction-following data, ignoring the unique challenge in long context alignment, i.e., selecting the samples enriched with meaningful long-range dependency relations.

By synthesizing long instruction-following data, Chen et al. (2024b); Bai et al. (2024) have effectively expanded the data volume for long context alignment. In this work, we aim to select influential samples from a vast ocean of synthetic data instead of indiscriminately increasing the quantity of data. Meanwhile, different from previous works (Li et al., 2024b; Xia et al., 2024) that only consider the selection of short instruction-following samples, we attempt to address the unique challenge in long context alignment, i.e., the necessity for modeling long-range dependencies. Thus, we propose **GATEAU** to measure the richness of long-range dependency relations in long samples. As shown in Figure 1, GATEAU consists of Homologous Models' Guidance and Contextual Awareness Measurement, which separately measure the difficulty of generating corresponding responses and understanding long input contexts due to the long-range dependencies.

## 2.1 HOMOLOGOUS MODELS' GUIDANCE

Modeling long-range dependencies is essential for long context alignment (Chen et al., 2024a). However, there is still no effective metric to directly quantify the richness of long-range dependency relations in data, which hinders the selection of influential data. Therefore, in this section, we attempt to approximately assess the richness of long-range dependency relations by measuring the difficulty in generating corresponding responses due to the long-range dependencies. If LLMs find it harder to generate target responses due to long-range dependencies, it means the sample has more complex and meaningful long-range dependency relations. An intuitive approach is to use the perplexity score to measure the difficulty of generating corresponding responses (Cao et al., 2024; Li et al., 2024b),

as the score evaluates the extent to which the LLM's output aligns with the corresponding correct answer. For a given long instruction-following sample $(c, x; y)$, the perplexity score of the given response $y$ from LLMs $\theta$ is calculated as:

$$\text{PPL}_\theta(y|c, x) = \text{Exp}(-\frac{1}{|y|} \sum_{i=1}^{|y|} \log P(y_i|c, x, y_{<i}; \theta)), \tag{1}$$

where $c$ means long input contexts and $x$ means the given instruction. A higher $\text{PPL}_\theta(y|c, x)$ indicates the harder the response of this long instruction-following data for LLM to generate.

However, we argue that a higher $\text{PPL}_\theta(y|x)$ does not mean the increased difficulty in generating corresponding responses is due to long-range dependencies. A higher $\text{PPL}_\theta(y|c, x)$ might be attributed to certain limited capabilities of LLMs, such as the limited instruction-following capability for the model without alignment, instead of handling the long-range dependency relations in this sample is more challenging for the LLM. Therefore, to minimize the influence of other factors, we propose **Homologous Models' Guidance (HMG)**. Specifically, we compare the perplexity scores of the response between two homologous models with different context windows to measure the difficulty due to the long-range dependencies. As homologous models (Yu et al., 2024) share the same pre-training stage and model architecture (e.g., LLaMA-2-7B-base-4k (Touvron et al., 2023) and LLaMA-2-7B-base-64k (Bai et al., 2024)), the only difference lies in their capabilities to model long-range dependency relations due to the extending context windows stage. Based on this motivation, we introduce the homologous models' perplexity score $\text{HMP}(c, x; y)$:

$$\text{HMP}(c, x; y) = \text{Norm}(\text{PPL}_{\theta_A}(y|c, x)) - \text{Norm}(\text{PPL}_{\theta_B}(y|c, x)). \tag{2}$$

Model $\theta_A$ employs short context windows and $\theta_B$ is the model with long ones, e.g., LLaMA-2-7B-base-4k $\theta_A$ and LLaMA-2-7B-base-64k $\theta_B$. We compute the difference in normalized perplexity scores between two homologous models with different context windows as the metric. We apply softmax normalization to each score to determine its respective ranking among the datasets, since perplexity scores of one sample from different models often can't be directly compared. By introducing a model $\theta_A$ with weaker long-range dependencies modeling capability but other similar capabilities learned during the pre-training stage, we mitigate the influence brought by lacking other capabilities compared to simply using perplexity score as Eq. (1). Thus, the difference in perplexity scores is primarily attributed to the different abilities in modeling long-range dependencies between model $\theta_A$ and model $\theta_B$. In other words, Eq. (2) reflects the difficulty of generating the corresponding response caused by long-range dependencies. We use the drop from $\text{PPL}_{\theta_A}$ to $\text{PPL}_{\theta_B}$ in Eq. (2) because model $\theta_A$ tends to produce a high perplexity score due to its weak ability to model long-range dependencies. Thus, a higher $\text{HMP}(c, x; y)$ indicates more difficulties for LLM in response generation due to the long-range dependencies, i.e., more long-range dependency relations in this sample.

## 2.2 CONTEXTUAL AWARENESS MEASUREMENT

Another challenge in long context alignment lies in enabling LLMs to understand and utilize the extremely long input contexts. Due to the long-range dependencies, it is hard for LLMs to utilize crucial information hidden in extremely long contexts, e.g., LLM's attention may focus on irrelevant content. Thus, we introduce **Contextual Awareness Measurement (CAM)** to evaluate whether LLMs' attention is appropriately focused on important segments within the long input contexts. Simply put, we attempt to evaluate the importance score of each segment and calculate the LLM's attention weights on each one, getting the **Contextual Awareness Score (CAS)** via calculating their similarity. For a given data $(c, x; y)$, we divide the input contexts $c$ into $N$ segments $[s_1, s_2, s_3, ..., s_N]$ of equal length $L$. Specifically, for a given segment $s_i$, we first compute the designed importance score $\text{IS}_\theta(s_i)$ and measure the significance of the segment in the response generation for LLM $\theta$:

$$\text{IS}_\theta(s_i) = \text{Norm}(\text{Exp}(-\frac{1}{|y|} \sum_{j=1}^{|y|} \log P(y_i|s_i, x, y_{<j}; \theta))). \tag{3}$$

We only keep the given segment $s_i$ as input contexts to calculate the perplexity score of generating the response $y$, indicating the difficulty of generating the corresponding response $y$ based on segment $s_i$. We apply softmax normalization to each score $\text{Exp}(-\frac{1}{|y|} \sum_{j=1}^{|y|} \log P(y_i|s_i, x, y_{<j}; \theta))$ to determine

its respective ranking among the segments $\{s_i\}_{i=1}^N$. Thus, the higher $\text{IS}_\theta(s_i)$ suggests a greater difficulty for LLM $\theta$ to generate the response based on segment $s_i$, implying that it is less important.

Once the importance scores of different segments are calculated, we then utilize the attention weights (i.e., the value of $\text{softmex}(\frac{QK^T}{\sqrt{d_k}})$ ) in the multi-head attention mechanism (Vaswani et al., 2017) to measure how the LLM utilizes these segments. To achieve it, we use the averaged attention weights of tokens $[t_1, ..., t_L]$ in segments $s_i$ as the score $\text{Attn}_\theta(s_i)$, which takes the form:

$$\text{Attn}_\theta(s_i) = \text{Norm}(\frac{1}{L} \sum_{j=1}^{L} \text{Attn}_\theta(t_j|y;\theta)), \tag{4}$$

where $\text{Attn}_\theta(t_j|y;\theta)$ means the attention weights averaged across the tokens in targeted response $y$ to the token $t_j$ in segment $s_i$. Meanwhile, we harness the attention weights averaged across different decoder layers and attention heads to thoroughly model how the LLM utilizes the long input contexts during the response generation (Hsieh et al., 2024). We apply softmax normalization to each score $\frac{1}{L} \sum_{j=1}^{L} \text{Attn}_\theta(t_j|y;\theta)$ to determine its respective ranking among the segments $\{s_i\}_{i=1}^N$ to yield the score $\text{Attn}_\theta(s_i)$. In so doing, we can calculate the attention weights between the response and segments, indicating how segments are utilized during the response generation.

Finally, we measure the difficulty of understanding the long input contexts due to long-range dependencies. For a given long instruction-following sample, we compute the CAS by resorting to the cosine similarity between importance scores $[\text{IS}_\theta(s_1), ..., \text{IS}_\theta(s_N)]$ and attention weights $[\text{Attn}_\theta(s_1), ..., \text{Attn}_\theta(s_N)]$, as follows:

$$\text{CAS}(c, x; y) = \text{CosSim}([\text{IS}_\theta(s_1), ..., \text{IS}_\theta(s_N)], [\text{Attn}_\theta(s_1), ..., \text{Attn}_\theta(s_N)]). \tag{5}$$

By doing this, we can measure the difficulty of understanding the long input contexts by checking whether LLMs' attention is focused on important segments. The insight is that should the LLM's attention focus more on less important segments, it suggests that the LLM struggles to accurately comprehend long input contexts. The higher $\text{CAS}(c, x; y)$ indicates more difficulties in utilizing the long input contexts to generate corresponding responses due to the long-range dependencies, which also implies the more long-range dependency relations in this sample.

## 2.3 SELECTING AND TRAINING

In practice, we frame the overall metric by weighting and summing both designed metrics to rank the data $(c, x; y)$, then select the most challenging samples as the influential samples, i.e.,

$$\text{Score}(c, x; y) = \alpha * \text{Norm}(\text{HMP}(c, x; y)) + (1 - \alpha) * \text{Norm}(\text{CAS}(c, x; y)). \tag{6}$$

$\alpha$ is a hyperparameter. We tap softmax normalization to the $\text{HMP}(c, x; y)$ and $\text{CAS}(c, x; y)$ of the given data across the whole dataset. Inspired by active learning (Li et al., 2024a), when trained on these challenging data characterized by complex long-range dependency relations, LLMs could effectively model the long-range dependencies and achieve better long context alignment.

LLMs are often fine-tuned with instruction-following data to learn to follow instructions. We aim to apply supervised fine-tuning on the selected data (e.g., selecting 10% samples of full datasets with top 10% scores according to Eq. (6)). Thus we train LLMs using the following objective function:

$$\mathcal{L}_\theta(c, x; y) = - \sum_{i=1}^{|y|} \log P(y_i|c, x, y_{<i};\theta). \tag{7}$$

It is similar to a language modeling loss, while only computing the loss associated with the response.

## 3 EXPERIMENT

### 3.1 EXPERIMENTAL SETUP

**Training Datasets.** We use LongAlign (Bai et al., 2024) as the long instruction-following dataset, which encompasses 10,000 long instruction-following samples. LongAlign is developed by using collected long sequences from 9 sources and applying the Self-Instruct (Wang et al., 2023b) approach

Table 1: Results (%) on LongBench in Real-world Settings. We use the ID to represent the dataset in LongBench, e.g., 1-1 is the ID of NarrativeQA dataset. More details can be found in Appendix C.2.

| Model | Single-Doc QA | | | | | Multi-Doc QA | | | | | Summarization | | | | |
|---|---|---|---|---|---|---|---|---|---|---|---|---|---|---|---|
| | 1-1 | 1-2 | 1-3 | 1-4 | Avg | 2-1 | 2-2 | 2-3 | 2-4 | Avg | 3-1 | 3-2 | 3-3 | 3-4 | Avg |
| **Auto Metrics** | | | | | | | | | | | | | | | |
| w/o SFT | 0.9 | 3.9 | 6.4 | 3.6 | 3.7 | 7.3 | 8.71 | 2.1 | 15.4 | 8.4 | 23.9 | 6.2 | 14.0 | 1.78 | 11.5 |
| w/o Long SFT | 16.8 | 29.1 | 45.8 | 48.7 | 35.1 | 27.8 | 17.6 | 11.4 | 25.3 | 20.5 | 27.4 | 23.3 | 27.8 | 14.3 | 23.2 |
| Full - 100% | 18.4 | 29.9 | 46.1 | **49.9** | 36.1 | 27.1 | 20.8 | 11.2 | 30.0 | 22.3 | 28.7 | 24.0 | 26.7 | 15.9 | 23.8 |
| Perplexity Guidance - 10% | 19.9 | 32.0 | 46.6 | 45.8 | 36.1 | 22.1 | 23.2 | 10.4 | 30.3 | 21.5 | 31.3 | 23.8 | 26.0 | 17.7 | 24.7 |
| CaR - 10% | 16.9 | 24.1 | 47.6 | 42.3 | 32.7 | 22.1 | 19.8 | 11.3 | 30.0 | 20.8 | **31.9** | 23.1 | 26.2 | 18.6 | 25.0 |
| Cherry Selection - 10% | 19.9 | 30.8 | 47.2 | 43.1 | 35.3 | 25.2 | 21.4 | 10.6 | 28.3 | 21.4 | 30.0 | 24.1 | 25.1 | 17.0 | 24.1 |
| **GATEAU-LLaMA - 10%** | **23.5** | **34.2** | **49.6** | **54.5** | **40.5** | **28.7** | **25.0** | **12.1** | **30.5** | **24.0** | 31.2 | **24.7** | **26.9** | **18.9** | **25.4** |
| Δ compared to Full - 100% | +5.1 | +4.3 | +3.5 | +4.6 | +4.4 | +1.6 | +4.2 | +0.9 | +0.5 | +1.8 | +2.5 | +0.7 | +0.2 | +3.0 | +1.6 |
| Perplexity Guidance - 30% | 21.1 | 33.6 | 46.1 | 46.7 | 36.9 | 23.4 | 21.0 | 10.1 | 30.1 | 21.2 | 30.2 | 24.7 | 26.4 | 18.9 | 25.1 |
| CaR - 30% | 18.0 | 24.4 | 46.9 | 45.0 | 33.6 | 25.4 | 20.8 | 14.4 | 29.4 | 22.5 | 30.1 | 24.8 | 26.5 | 18.2 | 24.9 |
| Cherry Selection - 30% | 20.5 | 33.1 | 48.0 | 51.0 | 38.2 | 26.7 | 20.4 | 13.5 | 29.1 | 22.4 | 30.4 | 24.1 | 26.9 | 17.7 | 24.8 |
| **GATEAU-LLaMA - 30%** | **23.7** | **34.1** | **49.6** | **54.6** | **40.5** | **30.1** | **23.8** | **14.9** | **30.4** | **24.8** | **30.5** | **24.9** | **27.2** | **18.9** | **25.4** |
| Δ compared to Full - 100% | +5.3 | +4.2 | +3.5 | +4.7 | +4.4 | +3.0 | +3.0 | +3.7 | +0.4 | +2.5 | +1.8 | +0.9 | +0.5 | +3.0 | +1.6 |
| Perplexity Guidance - 50% | 19.2 | 32.8 | 50.1 | 49.5 | 37.9 | 27.1 | 23.1 | 12.1 | 31.1 | 23.4 | **31.5** | 24.1 | 27.1 | 18.7 | 25.4 |
| CaR - 50% | 17.6 | 24.5 | 47.6 | 44.7 | 33.6 | 29.3 | 19.4 | 17.3 | 29.6 | 23.9 | 30.3 | 23.7 | 26.0 | 18.2 | 24.6 |
| Cherry Selection - 50% | 19.0 | 32.6 | 51.7 | 49.6 | 38.2 | 26.2 | 23.9 | 13.5 | 30.4 | 23.5 | 30.5 | 23.8 | 26.9 | 18.8 | 25.0 |
| **GATEAU-LLaMA - 50%** | **20.2** | **33.4** | **52.1** | 49.8 | **38.9** | **30.7** | **25.2** | **15.0** | **32.5** | **25.8** | 31.3 | **24.6** | **27.1** | 18.8 | **25.5** |
| Δ compared to Full - 100% | +1.8 | +3.5 | +6.0 | -0.1 | +2.8 | +3.6 | +4.4 | +3.8 | +2.5 | +3.6 | +2.6 | +0.6 | +0.4 | +2.9 | +1.6 |
| **GPT-4 Evaluation** | | | | | | | | | | | | | | | |
| w/o SFT | 33.8 | 38.0 | 41.1 | 34.8 | 36.9 | 41.3 | 37.2 | 33.3 | 42.0 | 38.5 | 39.2 | 20.2 | 37.1 | 30.9 | 31.9 |
| w/o Long SFT | 58.7 | 66.7 | 83.1 | 79.2 | 71.9 | 70.2 | 53.4 | 48.7 | 61.3 | 58.4 | 57.3 | 36.2 | 55.2 | 38.4 | 46.8 |
| Full - 100% | 62.8 | 69.0 | 83.1 | 81.3 | 74.1 | 71.5 | 54.8 | 51.3 | 66.2 | 61.0 | 58.7 | 39.8 | 57.6 | 41.2 | 49.3 |
| Perplexity Guidance - 10% | 62.0 | 68.8 | 86.4 | 85.6 | 75.7 | 73.5 | 59.7 | 52.1 | 68.2 | 63.4 | 67.6 | 41.3 | **67.0** | 44.9 | 55.2 |
| CaR - 10% | 60.3 | 69.0 | 86.0 | 84.8 | 75.0 | 69.1 | 58.3 | 52.3 | 68.5 | 62.1 | 64.1 | 41.4 | 60.3 | 42.1 | 52.0 |
| Cherry Selection - 10% | 60.8 | 67.2 | 86.7 | 84.3 | 74.8 | 71.3 | 57.8 | 51.0 | 69.0 | 62.3 | 61.3 | 40.0 | 64.8 | 41.5 | 51.9 |
| **GATEAU-LLaMA - 10%** | **63.6** | **69.2** | **86.9** | **87.1** | **76.7** | **74.8** | **60.8** | **53.1** | **69.5** | **64.6** | 67.6 | **42.6** | 66.2 | **47.8** | **56.1** |
| Δ compared to Full - 100% | +0.8 | +0.2 | +3.8 | +5.8 | +2.7 | +3.3 | +6.0 | +1.8 | +3.3 | +3.6 | +8.9 | +2.8 | +8.6 | +6.6 | +6.7 |
| Perplexity Guidance - 30% | 62.8 | 67.3 | 86.2 | 82.6 | 74.7 | 72.3 | 59.3 | 50.8 | 67.8 | 62.6 | 62.3 | 41.7 | 64.8 | 42.7 | 52.9 |
| CaR - 30% | 61.3 | 67.3 | 86.4 | 85.3 | 75.1 | 68.3 | 58.3 | 53.2 | 66.8 | 61.7 | 64.6 | 39.7 | 60.7 | 41.2 | 51.6 |
| Cherry Selection - 30% | 62.0 | 66.8 | 87.1 | 84.3 | 75.1 | 74.3 | 59.3 | 52.7 | 68.7 | 63.8 | 62.3 | 40.5 | 64.6 | 44.4 | 53.0 |
| **GATEAU-LLaMA - 30%** | **63.0** | **70.8** | **87.6** | **85.8** | **76.8** | **75.7** | **61.0** | **55.7** | **69.5** | **65.5** | **67.5** | **44.7** | **65.9** | **47.4** | **56.4** |
| Δ compared to Full - 100% | +0.2 | +1.8 | +4.5 | +4.5 | +2.8 | +4.2 | +6.2 | +4.4 | +3.3 | +4.5 | +8.8 | +4.9 | +8.3 | +6.2 | +7.1 |
| Perplexity Guidance - 50% | 63.1 | 68.1 | 87.8 | 82.1 | 75.3 | 74.2 | 59.2 | 52.5 | 69.2 | 63.8 | 64.7 | 41.1 | 65.7 | 42.1 | 53.4 |
| CaR - 50% | 60.0 | 66.3 | 85.6 | 84.2 | 74.0 | 70.7 | 55.8 | 54.3 | 68.2 | 62.3 | 64.4 | 41.1 | 60.8 | 40.3 | 51.7 |
| Cherry Selection - 50% | 62.8 | 65.5 | 86.2 | 82.8 | 74.3 | 72.2 | 56.8 | 52.7 | 67.8 | 62.4 | 64.6 | 39.4 | 64.1 | 42.1 | 52.6 |
| **GATEAU-LLaMA - 50%** | **63.5** | **70.3** | **89.7** | **86.5** | **77.5** | **75.3** | **60.8** | **53.5** | **68.5** | **64.5** | **65.1** | **41.6** | **65.9** | **46.1** | **54.7** |
| Δ compared to Full - 100% | +0.7 | +1.3 | +6.6 | +5.2 | +3.5 | +3.8 | +6.0 | +2.2 | +2.3 | +3.6 | +6.4 | +1.8 | +8.3 | +4.9 | +5.4 |

with long-context LLM Claude 2.1 (Anthropic., 2023). Though initially competitive, its dependence on Claude 2.1 synthesized data may lead to quality concerns. Thus, our method to apply the selection of long instruction data is based on the LongAlign dataset. Meanwhile, similar to Bai et al. (2024), to maintain the model's general capabilities and its proficiency in following short instructions, we utilize ShareGPT dataset (Chiang et al., 2023) as the source of short instruction data in our training data (empty assistant responses are filtered out). To further explore the effects of mixture proportions of long and short instruction-following samples, we evaluate our method in both **Real-world Settings** and **Limited Short Instruction Data Settings**. Real-world Settings (Bai et al., 2024) indicates real-world users prioritize short instruction-following interactions. Thus, to stay close to real-world situations, we attempt to use the full ShareGPT dataset as short instruction-following data. We also explore scenarios where short instruction data is limited, utilizing only the first 10% of ShareGPT, named Limited Short Instruction Data Settings.

**Training Settings.** In our experiments, we use LLaMA-2-7B-base-4k (Touvron et al., 2023) and LLaMA-2-7B-base-64k (Bai et al., 2024) as homologous models to apply the proposed Homologous Models' Guidance. LLaMA-2-7B-base-4k is a well-known open-sourced LLM with a context window

Table 2: Results (%) on LongBench-Chat in Real-world and Limited Short Instruction Data Settings.

| Model | Real-world | Limited |
|---|---|---|
| w/o SFT | 10.4 | 10.4 |
| w/o Long SFT | 37.4 | 36.2 |
| Full - 100% | 48.8 | 50.8 |
| Perplexity Guidance - 10% | 52.2 | 49.0 |
| CaR - 10% | 50.8 | 49.0 |
| Cherry Selection - 10% | 53.2 | 50.8 |
| **GATEAU-LLaMA - 10%** | **55.4** | **58.0** |
| Perplexity Guidance - 30% | 50.6 | 51.8 |
| CaR - 30% | 48.6 | 51.4 |
| Cherry Selection - 30% | 50.4 | 52.4 |
| **GATEAU-LLaMA - 30%** | **57.8** | **55.2** |
| Perplexity Guidance - 50% | 49.8 | 51.0 |
| CaR - 50% | 49.6 | 51.6 |
| Cherry Selection - 50% | 50.6 | 53.2 |
| **GATEAU-LLaMA - 50%** | **56.8** | **59.0** |

Table 3: Results (%) on LongBench in Limited Short Instruction Data Settings.

| Model | Single-Doc QA | | | | | Multi-Doc QA | | | | | Summarization | | | | |
|---|---|---|---|---|---|---|---|---|---|---|---|---|---|---|---|
| | 1-1 | 1-2 | 1-3 | 1-4 | Avg | 2-1 | 2-2 | 2-3 | 2-4 | Avg | 3-1 | 3-2 | 3-3 | 3-4 | Avg |
| **Auto Metrics** | | | | | | | | | | | | | | | |
| w/o SFT | 0.9 | 3.9 | 6.4 | 3.6 | 3.7 | 7.3 | 8.71 | 2.1 | 15.4 | 8.4 | 23.9 | 6.2 | 14.0 | 1.78 | 11.5 |
| w/o Long SFT | 13.8 | 19.2 | 38.3 | 37.1 | 27.1 | 15.2 | 14.7 | 8.2 | 25.7 | 16.0 | 29.4 | 24.4 | 25.0 | 19.3 | 24.5 |
| Full - 100% | 14.7 | 20.1 | 37.0 | 37.0 | 27.2 | 15.4 | 13.8 | 8.6 | 26.7 | 16.1 | 29.3 | 24.5 | 25.6 | 18.6 | 24.5 |
| Perplexity Guidance - 10% | 15.4 | 19.2 | 41.0 | 37.8 | 28.4 | 15.0 | 14.8 | **8.5** | 25.6 | 16.0 | 28.8 | 23.9 | 26.1 | 17.8 | 24.2 |
| CaR - 10% | 11.5 | 17.7 | 37.7 | 30.0 | 24.2 | 15.6 | 12.5 | 8.4 | 25.9 | 15.6 | 29.3 | 24.1 | 26.2 | 18.2 | 24.5 |
| Cherry Selection - 10% | 14.6 | 19.2 | 41.2 | 37.7 | 28.2 | 15.7 | 14.6 | 7.6 | 25.3 | 15.8 | 29.4 | 24.1 | 26.0 | 17.8 | 24.3 |
| **GATEAU-LLaMA - 10%** | **17.1** | **20.7** | **43.4** | **38.3** | **29.9** | **19.9** | **18.5** | 8.2 | **26.8** | **18.4** | **29.6** | **24.3** | **26.3** | **18.3** | **24.6** |
| Δ compared to Full - 100% | +2.4 | +0.6 | +6.4 | +1.3 | +2.7 | +4.5 | +4.7 | -0.4 | +0.1 | +2.2 | +0.3 | -0.2 | +0.7 | -0.3 | +0.1 |
| Perplexity Guidance - 30% | 15.3 | **20.6** | 42.3 | 38.2 | 29.1 | 17.4 | 15.9 | 8.6 | 27.5 | 17.4 | 28.3 | 24.3 | 25.7 | 19.0 | 24.3 |
| CaR - 30% | 13.6 | 18.3 | 41.0 | 30.5 | 25.9 | 16.7 | 15.8 | 9.4 | 27.0 | 17.2 | 28.8 | 24.3 | 25.3 | 18.4 | 24.2 |
| Cherry Selection - 30% | 15.9 | 19.5 | 42.3 | **39.0** | 29.2 | 17.3 | 16.3 | 9.3 | 26.2 | 17.3 | 29.2 | **25.0** | 26.1 | 18.2 | 24.6 |
| **GATEAU-LLaMA - 30%** | **17.7** | 20.4 | **43.1** | 38.6 | **29.9** | **22.5** | **18.5** | **11.6** | **27.7** | **20.1** | **30.5** | 24.3 | **26.8** | **19.7** | **25.3** |
| Δ compared to Full - 100% | +3.0 | +0.3 | +6.1 | +1.6 | +2.7 | +7.1 | +4.7 | +3.0 | +1.0 | +4.0 | +1.2 | -0.2 | +1.2 | +1.1 | +0.8 |
| Perplexity Guidance - 50% | 16.4 | 20.6 | 39.1 | 37.1 | 28.3 | 16.7 | 16.4 | 8.2 | 26.0 | 16.8 | 29.3 | 25.1 | 25.2 | 19.1 | 24.7 |
| CaR - 50% | 12.1 | 18.1 | 40.4 | 30.4 | 25.3 | 17.3 | 15.1 | 9.0 | 26.3 | 16.9 | 28.3 | 23.6 | 25.1 | 18.9 | 24.0 |
| Cherry Selection - 50% | 15.5 | 19.5 | 38.9 | 37.3 | 27.8 | 15.4 | 16.3 | 8.8 | 26.1 | 16.7 | **30.6** | 24.8 | 25.3 | 18.9 | 24.9 |
| **GATEAU-LLaMA - 50%** | **18.5** | **22.5** | **43.9** | **39.1** | **31.0** | **17.9** | **16.7** | **9.6** | **28.0** | **18.1** | 30.1 | **25.3** | **26.6** | **19.4** | **25.3** |
| Δ compared to Full - 100% | +3.8 | +2.4 | +6.9 | +2.1 | +3.8 | +2.5 | +2.9 | +1.0 | +1.3 | +1.9 | +0.8 | +0.8 | +0.9 | +0.8 | +0.8 |
| **GPT-4 Evaluation** | | | | | | | | | | | | | | | |
| w/o SFT | 33.8 | 38.0 | 41.1 | 34.8 | 36.9 | 41.3 | 37.2 | 33.3 | 42.0 | 38.5 | 39.2 | 20.2 | 37.1 | 30.9 | 31.9 |
| w/o Long SFT | 62.3 | 70.8 | 88.5 | 82.7 | 76.1 | 72.8 | 60.6 | 51.8 | 67.3 | 63.1 | 64.7 | 41.1 | 61.4 | 41.6 | 52.2 |
| Full - 100% | 58.7 | 69.7 | 85.8 | 83.0 | 74.3 | 70.5 | 58.7 | 50.8 | 67.8 | 62.0 | 59.6 | 38.4 | 59.6 | 43.3 | 50.2 |
| Perplexity Guidance - 10% | 62.8 | 69.2 | 89.3 | 85.7 | 76.8 | 73.8 | 59.1 | 54.1 | **71.1** | 64.5 | 69.8 | 45.8 | 65.7 | 50.1 | 57.9 |
| CaR - 10% | 62.8 | 68.3 | 88.0 | 82.7 | 75.5 | 71.8 | 58.0 | 52.7 | 68.8 | 62.8 | 65.5 | 42.0 | 61.8 | 43.1 | 53.1 |
| Cherry Selection - 10% | 62.8 | 69.8 | 86.7 | 85.7 | 76.3 | 72.0 | 58.7 | 52.5 | 69.3 | 63.1 | 63.2 | 43.3 | 60.1 | 46.4 | 53.3 |
| **GATEAU-LLaMA - 10%** | **64.8** | **74.7** | **89.8** | **86.5** | **79.0** | **75.2** | **61.2** | **54.6** | 70.0 | **65.3** | **71.1** | **47.3** | **67.0** | **54.2** | **59.9** |
| Δ compared to Full - 100% | +6.1 | +5.0 | +4.0 | +3.5 | +4.7 | +4.7 | +2.5 | +3.8 | +2.2 | +3.3 | +11.5 | +8.9 | +7.4 | +10.9 | +9.7 |
| Perplexity Guidance - 30% | 62.5 | 71.8 | 88.2 | 83.8 | 76.6 | 74.6 | 58.5 | 53.5 | 69.3 | 64.0 | 67.5 | 44.0 | 64.7 | 50.4 | 56.7 |
| CaR - 30% | 60.8 | 70.7 | 88.4 | 81.8 | 75.4 | 73.0 | 59.0 | 53.5 | 68.5 | 63.5 | 64.1 | 40.9 | 62.3 | 45.8 | 53.3 |
| Cherry Selection - 30% | 62.8 | 71.7 | 88.9 | **87.5** | 77.7 | 70.3 | 58.7 | 50.3 | 68.2 | 61.9 | 62.9 | 43.5 | 65.2 | 44.6 | 54.1 |
| **GATEAU-LLaMA - 30%** | **64.8** | **73.0** | **89.3** | 86.2 | **78.3** | **74.7** | **61.0** | **54.2** | **69.8** | **64.9** | **70.8** | **46.0** | **66.4** | **51.4** | **58.7** |
| Δ compared to Full - 100% | +6.1 | +3.3 | +3.5 | +3.2 | +4.0 | +4.2 | +2.3 | +3.4 | +2.0 | +3.0 | +11.2 | +7.6 | +6.8 | +8.1 | +8.4 |
| Perplexity Guidance - 50% | 61.5 | 68.3 | 85.1 | 82.8 | 74.4 | 72.3 | 59.3 | 52.0 | 67.7 | 62.8 | 60.2 | 40.9 | 58.6 | 42.3 | 50.5 |
| CaR - 50% | 62.3 | 68.1 | 86.9 | 80.1 | 74.4 | 71.0 | 58.7 | 52.8 | 68.0 | 62.6 | 64.4 | 41.2 | 61.1 | 45.6 | 53.1 |
| Cherry Selection - 50% | 61.2 | 69.7 | 86.2 | 83.7 | 75.2 | 69.7 | 56.8 | 49.5 | 66.2 | 60.6 | 64.1 | 41.8 | 60.5 | 43.7 | 52.5 |
| **GATEAU-LLaMA - 50%** | **63.7** | **71.8** | **87.1** | **84.7** | **76.8** | **74.0** | **60.0** | **53.8** | **69.0** | **64.2** | **66.1** | **43.9** | **62.4** | **46.4** | **54.7** |
| Δ compared to Full - 100% | +5.0 | +2.1 | +1.3 | +1.7 | +2.5 | +3.5 | +1.3 | +3.0 | +1.2 | +2.3 | +6.5 | +5.5 | +2.8 | +3.1 | +4.5 |

of 4k tokens. To extend context windows of LLaMA-2, Bai et al. (2024) propose LLaMA-2-7B-base-64k by modifying the RoPE position encoding (Su et al., 2023) and applying continual training on data with lengths under 64k, for a total of 10 billion tokens. Meanwhile, for LLaMA-2-7B-base-4k, we expand the base frequency $b$ of the RoPE position encoding by 200 times (from 10,000 to 2,000,000) to extend the context windows and avoid the model conducting extreme perplexity score (>1,000) in Homologous Models' Guidance. For Contextual Awareness Measurement, we use LLaMA-2-7B-base-64k to calculate the score as we use selected samples to train the LLaMA-2-7B-base-64k as our final model **GATEAU-LLaMA**. We also use 13B-scale LLaMA (i.e., LLaMA-2-13B-base-4k (Touvron et al., 2023) and LLaMA-2-13B-base-64k (Bai et al., 2024)) to explore whether our method fits larger LLMs. More details are shown in the Appendix A.

Table 4: Results (%) on MT-Bench in both Real-world and Limited Short Instruction Data Settings.

| Model | Real-world | Limited |
|---|---|---|
| w/o SFT | 34.6 | 34.6 |
| w/o Long SFT | 53.7 | 50.5 |
| Full - 100% | 54.3 | 47.7 |
| Perplexity Guidance - 10% | 56.1 | 50.9 |
| CaR - 10% | 54.9 | 49.9 |
| Cherry Selection - 10% | 56.8 | 47.6 |
| **GATEAU-LLaMA - 10%** | **58.6** | **53.4** |
| Perplexity Guidance - 30% | 55.0 | 50.2 |
| CaR - 30% | 54.3 | 48.6 |
| Cherry Selection - 30% | 54.3 | 45.8 |
| **GATEAU-LLaMA - 30%** | **58.8** | **52.9** |
| Perplexity Guidance - 50% | 55.9 | 49.2 |
| CaR - 50% | 54.7 | 51.2 |
| Cherry Selection - 50% | 56.3 | 49.6 |
| **GATEAU-LLaMA - 50%** | **57.3** | **54.2** |

**Baselines.** We compare our method **GATEAU** with multiple instruction data selection baselines, including variants of our proposed method and methods that focus on the selection of short instruction data. **Cherry Selection** (Li et al., 2024b) and **CaR** (Ge et al., 2024) are state-of-the-art methods to select the influential short instruction-following data. We also use the perplexity score from long-context LLM as guidance to select long instruction-following samples according to Eq. (1), named as **Perplexity Guidance**. More details can be found in the Appendix B.

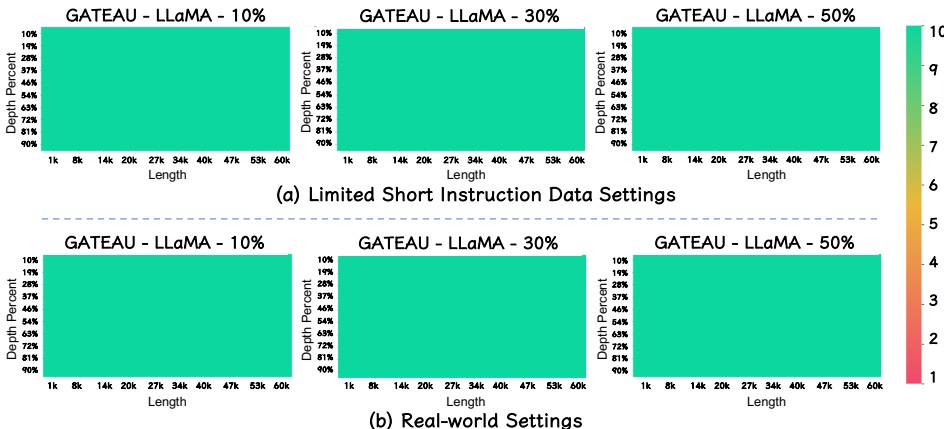

Figure 2: Needle in the Haystack test.

**Evaluation.** To gauge the effectiveness of our method, we conduct extensive evaluations on different benchmarks. We use **LongBench-Chat** (Bai et al., 2024) to evaluate the models' long context alignment proficiency, which is a benchmark that compromises open-ended questions of 10k-100k in length. It covers diverse aspects of instruction-following abilities such as reasoning, coding, summarization, and multilingual translation over long contexts. GPT-4 (OpenAI, 2023) is employed to score the machine-generated responses based on the annotated ground-truths and few-shot scoring examples. We also employ bilingual and multi-task benchmark **LongBench** (Bai et al., 2023) to evaluate the model's long context understanding abilities. We conduct evaluations on three types of tasks the same as Bai et al. (2024), including Single-Doc QA, Multi-Doc QA, and Summarization. Meanwhile, as aligned models generally produce longer responses, rather than relying solely on the original automated metrics (e.g., ROUGE, F1) to evaluate the models' replies, we keep the same as Bai et al. (2024) to employ GPT-4 to evaluate the model outputs based on their alignment with the ground-truth answers on LongBench. We use **MT-Bench** (Zheng et al., 2023), a multi-turn chat benchmark, to measure the models' ability to follow short instructions via GPT-4 rating. To ensure the most stable evaluation results, we use GPT-4 to score twice on LongBench-Chat, MT-Bench, and LongBench, and average these scores to obtain the final score. More details about evaluation (e.g., the rating prompts) can be found in Appendix C.

### 3.2 IMPACT OF GATEAU

**Enhancing the Long-Context Understanding Capabilities.** The experimental results are shown in Table 1 and Table 3 for the LongBench benchmark. Our methods achieve consistent and remarkable performance gains in both different settings and evaluations. We show the improved scores (Δ *compared to Full-100%*) compared to indiscriminately using the whole dataset (*Full-100%*), indicating that GATEAU helps LLM to better understand and utilize the long input contexts. Further, we find that the baselines focusing on the selection of short instruction-following data (*CaR* and *Cherry Selection*) hold inferior results, sometimes even worse than indiscriminately using the whole dataset (*Full-100%*). This can be attributed to these methods are not designed for long context alignment and understanding, thus failing to select the samples enriched with long-range dependency relations. Meanwhile, we can see that using 30% of the whole long instruction-following dataset (*GATEAU-LLaMA-30%*) can achieve the best performance of LongBench in two different settings. This is because its ability to maintain an optimal balance between the volume and quality of the long instruction-following samples it utilizes, leading to the most desirable results.

**Improving Instruction-Following Capabilities for Both Short and Long Inputs.** The experimental results are presented in Table 2 and Table 4 for the LongBench-Chat and MT-Bench benchmarks in two settings. It shows our proposed method GATEAU can consistently improve LLMs' capabilities in following both long and short instructions and generating high-quality responses. Compared to indiscriminately using the whole dataset (*Full-100%*), using the selected subset of the long instruction-following dataset (*GATEAU-LLaMA*) can significantly improve the instruction-following capabilities, e.g., increasing 9% in LongBench-Chat and 6.5% in MT-Bench. Meanwhile, the low performance of *w/o Long SFT* in LongBench-Chat indicates that using long instruction-following data is important for the performance of LLMs in handling the instructions with long input contexts. The results also

Table 5: Results (%) of ablation study and scalability test.

| Model | LongBench | | | LongBench-Chat | MT-Bench | | |
|---|---|---|---|---|---|---|---|
| | Single-Doc QA | Multi-Doc QA | Summarization | Avg | First-turn | Second-turn | Avg |
| **Real-world Settings** | | | | | | | |
| **GATEAU-LLaMA - 13B - 50%** | **40.2** | **27.1** | **25.7** | **61.4** | **66.8** | **55.3** | **61.1** |
| -w/o Contextual Awareness Measurement | 38.1 | 25.8 | 24.6 | 60.2 | 66.2 | 55.0 | 60.6 |
| -w/o Homologous Models' Guidance | 38.6 | 26.0 | 25.1 | 60.6 | 66.0 | 54.6 | 60.3 |
| -w/o Data Selection (i.e., Full - 100%) | 33.6 | 16.7 | 24.4 | 59.4 | 66.0 | 54.1 | 59.6 |
| **GATEAU-LLaMA - 7B - 50%** | **38.9** | **25.8** | **25.5** | **56.8** | **64.1** | **50.4** | **57.3** |
| -w/o Contextual Awareness Measurement | 38.4 | 24.3 | 25.1 | 53.2 | 61.7 | 51.5 | 56.6 |
| -w/o Homologous Models' Guidance | 38.6 | 24.5 | 24.9 | 52.8 | 63.1 | 49.3 | 56.3 |
| -w/o Data Selection (i.e., Full - 100%) | 36.1 | 22.3 | 23.8 | 48.8 | 60.0 | 48.7 | 54.3 |
| **Limited Short Instruction Data Settings** | | | | | | | |
| **GATEAU-LLaMA - 13B - 50%** | **32.1** | **19.1** | **25.3** | **62.6** | **66.0** | **51.5** | **58.8** |
| -w/o Contextual Awareness Measurement | 31.4 | 18.4 | 24.7 | 59.6 | 64.2 | 50.3 | 57.3 |
| -w/o Homologous Models' Guidance | 30.8 | 18.6 | 25.0 | 60.4 | 63.6 | 50.6 | 57.1 |
| -w/o Data Selection (i.e., Full - 100%) | 30.4 | 17.8 | 24.5 | 54.2 | 61.0 | 49.8 | 55.4 |
| **GATEAU-LLaMA - 7B - 50%** | **31.0** | **18.1** | **25.3** | **59.0** | **64.2** | **44.1** | **54.2** |
| -w/o Contextual Awareness Measurement | 28.5 | 17.5 | 24.7 | 53.2 | 61.3 | 42.4 | 51.8 |
| -w/o Homologous Models' Guidance | 28.7 | 17.3 | 24.6 | 54.4 | 56.1 | 45.0 | 50.6 |
| -w/o Data Selection (i.e., Full - 100%) | 27.2 | 16.1 | 24.5 | 50.8 | 54.5 | 40.9 | 47.7 |

show that our method GATEAU achieves uniformly better performance in varying ratios of used long instruction-following samples compared with other baselines, indicating the effectiveness of our method. Compared with baselines focusing on short instruction-following samples (*CaR* and *Cherry Selection*), GATEAU can identify samples enriched with long-range dependency relations more effectively and help LLMs to achieve better overall performance. Also, we observe that the selection of long instruction-following samples aids in augmenting the instruction-following capabilities for short inputs. We conjecture that handling complex tasks (i.e., long input contexts) contributes to handling the easy ones (i.e., short input contexts).

## 3.3 DISCUSSION

**Needle in the Haystack Test.** We conduct the "Needle in A HayStack" experiment (result visualization in Figure 2 ) to test the model's ability to utilize information from 10 different positions within long contexts of varying lengths between 1k-60k. Specifically, this task asks for the model to retrieve a piece of fact (the 'needle') that is inserted in the middle (positioned at a specified depth percent) of a long context window (the 'haystack'). These results show that GATEAU can help LLM's ability to utilize information from different positions within long texts, resulting in a decrease in the model's retrieval error.

**Ablation Study.** To evaluate the effectiveness of two designed metrics, including Homologous Models' Guidance and Contextual Awareness Measurement, we conduct the ablation study in Table 5. One can observe that Homologous Models' Guidance and Contextual Awareness

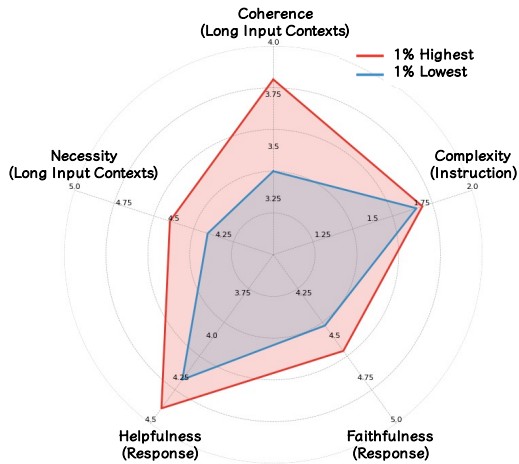

Figure 3: The comparison between samples with top 1% and least 1% scored by our method.

Measurement can both enhance LLMs' instruction-following and long-context understanding capabilities. This indicates the effectiveness of GATEAU and using both two methods can further improve the overall performance as they separately measure the difficulty of generating corresponding responses and understanding long input contexts due to the long-range dependencies.

**Scalability Test.** We explore whether our method GATEAU can fit in larger LLMs in Table 5. To do so, we apply GATEAU on Llama-2-13B and fine-tune Llama-2-13B-64k (Bai et al., 2024) using the selected samples. Compared to the 7B-scale model (*GATEAU-LLaMA-7B*), the 13B model (*GATEAU-LLaMA-13B*) shows consistent improvements on three benchmarks. This indicates that GATEAU scales effectively to larger-scale models.

**General Characteristics of Selected Samples.** We delve into whether the selected samples based on our method align with known characteristics of high-quality training data as shown in Figure 3. To this end, we select 100 samples with the top 1% scores and 100 samples with the least 1% scores.

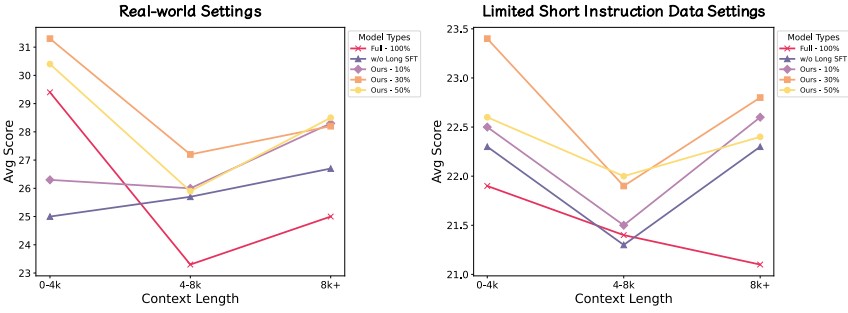

Figure 4: Average score (%) under different context lengths on LongBench.

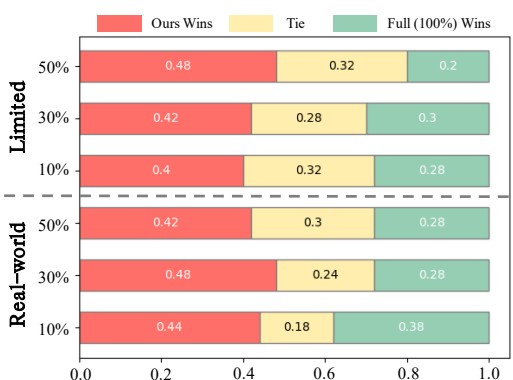

Figure 5: Human evaluation in two settings.

Utilizing GPT-4, we evaluate each sample on five aspects: the coherence of long input contexts, the necessity of long input contexts, helpfulness of response, the faithfulness of response, and the complexity of instruction. A sample with a higher score tends to be more high-quality, especially the long input contexts and the response of the sample. It also illustrates the difference between samples with high or low scores and verifies the effectiveness of GATEAU in identifying the influential samples. The complexity of instruction, in particular, shows a mere improvement compared to other characteristics. We further evaluate the whole dataset on this characteristic and find that all samples show consistently low scores, which may be due to the limitation of the synthetic dataset. As these samples are synthesized by close-source LLMs, the instructions are easy for these close-source LLMs. More details can be found in the Appendix D.1.

**Human Evaluation.** To better illustrate the efficacy of our method, further human evaluation is conducted. Specifically, we evaluate the whole LongBench-Chat benchmark, which consists of 50 instances. We invited three human participants (all of them are Ph.D. students or Master students) to compare the responses generated by the models. For each comparison, three options are given (Win, Tie, and Loss) and the final results are determined by the majority voting of the participants. Table 5 showcases the effectiveness of our method, i.e., our trained models show consistent preference from human participants. More details can be found in the Appendix D.4.

**Variation of Abilities under Different Context Lengths.** Figure 4 reports the macro-average scores (%) on data in length ranges of 0-4k, 4k-8k, and 8k+. We can find that our method improves the performance in long input contexts scenarios (i.e., 4k-8k and 8k+) compared to using the whole training dataset (*Full-100%*). Meanwhile, indiscriminately utilizing the whole long SFT dataset (*Full-100%*) even hinders the performance in long input contexts scenarios (i.e., 4k-8k and 8k+) compared to only utilizing short instruction-following dataset (*-w/o Long SFT*). This further confirms the necessity of selecting influential samples and the effectiveness of our method.

## 4 CONCLUSION

In this study, we introduce **GATEAU**, a new novel framework designed to select influential samples for long context alignment. Different from previous studies for selecting the short SFT samples, we attempt to address the unique challenge in long context alignment, i.e., the necessity for modeling long-range dependencies. To measure the richness of long-range dependency relations in long SFT samples, we propose Homologous Models' Guidance and Contextual Awareness Measurement to separately measure the difficulty of generating corresponding responses and understanding long input contexts due to the long-range dependencies. Trained on these selected influential samples based on our method, our model achieves better alignment. Extensive experimental evaluation and analysis have consistently shown the effectiveness of our proposed GATEAU compared to other methods.

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

## A    TRAINING DETAILS

All models are trained with 8xA800 80G GPUs and DeepSpeed+ZeRO3+CPU offloading. We use BF16 in both our training and inference. The models can be trained with a maximum length of 64k tokens without GPU memory overflow. Consequently, we set the maximum length of the training data to 64k, with any data exceeding this length being truncated from the right side. We keep the same maximum length in the Homologous Model's Guidance and Contextual Awareness Measurement but truncated from the left side to keep the original responses. We set the batch size to 8, with a gradient accumulation step of 12 for all the training methods. We train 2 epochs on the training data. We set the learning rate as 2e-5 and use AdamW (Loshchilov & Hutter, 2019) as our optimizer. The $\beta_1$ and $\beta_2$ in AdamW optimizer are set to 0.9 and 0.95. Meanwhile, the length of segment $L$ is set to 128 in Contextual Awareness Measurement. Hyperparameter $\alpha$ in Eq. (6) is set to 0.7 in Limited Short Instruction Data settings and 0.8 in Real-world Settings.

## B    BASELINES

In this section, we detail the design of baselines in our experiments.

**w/o SFT.**    For w/o SFT, we directly utilize the base model without alignment to get the experiment results, i.e., the results of LLaMA-2-7B-base-64k.

**w/o Long SFT.** For w/o Long SFT, we just use the short instruction data from ShareGPT to apply the supervised fine-tuning stage for alignment. The number of used short instruction samples from ShareGPT is determined by the different settings.

**Full - 100%.** For Full - 100%, we use the full data of LongAlign, including 10k long instruction samples, to conduct the supervised fine-tuning for alignment. The number of used short instruction samples from ShareGPT is determined by the different settings.

**Perplexity Guidance.** We use the perplexity score from LLM as guidance to select long instruction-following samples according to Eq. (1). We select the long instruction-following samples with the highest perplexity scores as the most influential samples to train the model. Meanwhile, the number of used short instruction samples from ShareGPT is determined by the different settings.

**CaR.** This work (Ge et al., 2024) proposes a straightforward yet efficacious short instruction-following selection framework. This method first selects a subset that ensures the retention of a large number of high-quality instructions and then supplements a small number of high-quality instructions from each cluster to enhance the diversity of the data while preserving instruction quality. Specifically, this work first employs a small-scale trained reward model (355M parameters) to get the score of the samples. Meanwhile, the cluster model is employed to cluster all candidate instruction pairs into $k$ clusters Finally, all instruction pairs are sorted based on their scores, and the top $n_1$ pairs are selected; within each cluster, instruction pairs are sorted by score, and the top $n_2$ pairs are chosen. A high-quality sub-dataset with preserved diversity is then curated by duplicating $n_1 + k \times n_2$ pairs of instructions. We directly use the same reward model and hyperparameters to select long instruction-following samples. Meanwhile, the number of used short instruction samples from ShareGPT is determined by the different settings.

**Cherry Selection.** Li et al. (2024b) proposes a method for autonomously sifting through expansive open-source short instruction-following datasets to discover the most influential training samples. At the heart of this method is the hypothesis that during their preliminary training stages with carefully chosen instruction data, LLMs can develop an intrinsic capability to discern instructions. This foundational understanding equips them with the discernment to assess the quality of broader datasets thus making it possible to estimate the instruction-following difficulty in a self-guided manner. To estimate the difficulty of a given example, this work proposes a novel metric called Instruction-Following Difficulty (IFD) score in which both models' capability to generate a response to a given instruction and the models' capability to generate a response directly are measured and compared. By calculating IFD scores, this method quantifies the challenge each sample presents to the model and utilizes selected data with standout IFD scores to hone the model. We apply this method to select the long instruction-following samples as the baseline. Meanwhile, the number of used short instruction samples from ShareGPT is determined by the different settings.

## C EVALUATIONS

### C.1 LONGBENCH-CHAT

**Evaluation Data.** LongBench-Chat focuses on assessing LLMs' instruction-following capability under the long context. LongBench-Chat includes 50 long context real-world queries ranging from 10k to 100k in length, covering various key user-intensive scenarios such as document QA, summarization, and coding. It consists of 40 tasks in English and 10 in Chinese.

**Evaluation Prompts.** LongBench-Chat employs GPT-4 to score the model's response in 1-10 based on a given human-annotated referenced answer and few-shot scoring examples for each question. We use the same prompt as LongBench-Chat to get GPT-4's evaluation:

> **LongBench-Chat Evaluation Prompt**
>
> [Instructions] You are asked to evaluate the quality of the AI assistant's answers to user questions as an impartial judge, and your evaluation should take into account factors including correctness (high priority), helpfulness, accuracy, and relevance. The scoring principles are as follows:
> 1. Read the AI assistant's answer and compare the assistant's answer with the reference answer.
> 2. Identify all errors in the AI Assistant's answers and consider how much they affect the answer to the question.
> 3. Evaluate how helpful the AI assistant's answers are in directly answering the user's questions and providing the information the user needs.
> 4. Examine any additional information in the AI assistant's answer to ensure that it is correct and closely related to the question. If this information is incorrect or not relevant to the question, points should be deducted from the overall score.
> Please give an overall integer rating from 1 to 10 based on the above principles, strictly in the following format:"[[rating]]", e.g. "[[5]]".
>
> [Question] { }
> [Reference answer begins] { } [Reference answer ends]
> Below are several assistants' answers and their ratings:
> [Assistant's answer begins] { } [Assistant's answer ends]
> Rating: [[{ }]]
> [Assistant's answer begins] { } [Assistant's answer ends]
> Rating: [[{ }]]
> [Assistant's answer begins] { } [Assistant's answer ends]
> Rating: [[{ }]]
>
> Please rate the following assistant answers based on the scoring principles and examples above:
> [Assistant's answer begins] { } [Assistant's answer ends]
> Rating:

### C.2 LONGBENCH

**Evaluation Data.** LongBench is the first bilingual, multitask benchmark tailored for long context understanding. LongBench includes different languages (Chinese and English) to provide a more comprehensive evaluation of the large models' bilingual capabilities in long-context understanding. Detailed statistics of the used dataset in LongBench can be found in Table 6.

**Evaluation Prompts.** We also conduct GPT-4 evaluation for LongBench. As aligned models generally produce longer responses, rather than relying solely on the original automated metrics (ROUGE, F1) to evaluate the models' replies, we additionally employ GPT-4 to assess the model outputs based on their alignment with the ground-truth answers on LongBench. For the first two QA tasks, the prompt for the GPT-4 evaluator is the same as Bai et al. (2024):

Table 6: An overview of the dataset statistics in LongBench. 'Source' denotes the origin of the context. 'Avg len' (average length) is computed using the number of words for the English datasets and the number of characters for the Chinese datasets.

| Dataset | ID | Source | Avg len | Auto Metric | Language | #data |
|---|---|---|---|---|---|---|
| *Single-Document QA* | | | | | | |
| NarrativeQA | 1-1 | Literature, Film | 18,409 | F1 | English | 200 |
| Qasper | 1-2 | Science | 3,619 | F1 | English | 200 |
| MultiFieldQA-en | 1-3 | Multi-field | 4,559 | F1 | English | 150 |
| MultiFieldQA-zh | 1-4 | Multi-field | 6,701 | F1 | Chinese | 200 |
| *Multi-Document QA* | | | | | | |
| HotpotQA | 2-1 | Wikipedia | 9,151 | F1 | English | 200 |
| 2WikiMultihopQA | 2-2 | Wikipedia | 4,887 | F1 | English | 200 |
| MuSiQue | 2-3 | Wikipedia | 11,214 | F1 | English | 200 |
| DuReader | 2-4 | Baidu Search | 15,768 | Rouge-L | Chinese | 200 |
| *Summarization* | | | | | | |
| GovReport | 3-1 | Government report | 8,734 | Rouge-L | English | 200 |
| QMSum | 3-2 | Meeting | 10,614 | Rouge-L | English | 200 |
| MultiNews | 3-3 | News | 2,113 | Rouge-L | English | 200 |
| VCSUM | 3-4 | Meeting | 15,380 | Rouge-L | Chinese | 200 |

---

**LongBench Evaluation Prompt for QA tasks**

You are asked to evaluate the quality of the AI assistant's answers to user question as an impartial judge, and your evaluation should take into account factors including correctness (high priority), and comprehensiveness (whether the assistant's answer covers all points). Read the AI assistant's answer and compare against the reference answer, and give an overall integer rating in 1, 2, 3 (1 = wrong or irrelevant, 2 = partially correct, 3 = correct and comprehensive) based on the above principles, strictly in the following format:"[[rating]]", e.g. "[[2]]".

Question: {*Question*}
Reference answer: {*Groundtruth*}
Assistant's answer: {*Response*}
Rating:

---

The prompt for GPT-4 evaluation on summarization tasks is the same as Bai et al. (2024):

**LongBench Evaluation Prompt for summarization tasks**

You are asked to evaluate the quality of the AI assistant's generated summary as an impartial judge, and your evaluation should take into account factors including correctness (high priority), comprehensiveness (whether the assistant's summary covers all points), and coherence. Read the AI assistant's summary and compare against the reference summary, and give an overall integer rating in on a scale of 1 to 5, where 1 is the lowest and 5 is the highest based on the evaluation criteria, strictly in the following format:"[[rating]]", e.g. "[[3]]".

Reference summary: {*Groundtruth*}
Assistant's summary: {*Response*}
Rating:

---

## C.3 MT-BENCH

**Evaluation Data.** MT-Bench is a benchmark consisting of 80 high-quality multi-turn questions. MT-bench is designed to test multi-turn conversation and instruction-following ability, covering common use cases and focusing on challenging questions to differentiate models. MT-Bench is also carefully constructed to differentiate chatbots based on their core capabilities, including writing, roleplay, extraction, reasoning, math, coding, knowledge I (STEM), and knowledge II (humanities/social science). To automate the evaluation process, MT-Bench prompts strong LLMs like GPT-4 to act as judges and assess the quality of the models' responses. In MT-bench, we use single-answer grading

Table 7: Detailed results (%) of MT-Bench.

| Model | First-turn | Second-turn | Writing | Roleplay | Reasoning | Math | Coding | Extraction | STEM | Humanities |
|---|---|---|---|---|---|---|---|---|---|---|
| **Real-world Settings** | | | | | | | | | | |
| w/o SFT | 43.5 | 25.6 | 44.5 | 44.0 | 35.0 | 16.5 | 18.0 | 28.0 | 42.0 | 48.8 |
| w/o Long SFT | 60.0 | 47.4 | 73.8 | 72.0 | 44.0 | 22.0 | 25.5 | 42.5 | 63.0 | 86.5 |
| Full - 100% | 60.0 | 48.7 | 78.5 | 70.3 | 45.5 | 19.0 | 29.0 | 42.0 | 67.5 | 83.0 |
| Perplexity Guidance - 10% | 63.1 | 48.9 | 68.7 | 67.0 | 43.5 | 26.5 | 33.2 | 50.5 | 69.8 | 88.5 |
| CaR - 10% | 59.8 | 50.0 | 76.5 | 75.3 | 44.5 | 24.5 | 24.8 | 43.5 | 64.2 | 84.9 |
| Cherry Selection - 10% | 63.0 | 50.5 | 74.5 | 73.8 | 42.3 | 25.0 | 32.5 | 48.3 | 70.3 | 87.5 |
| **GATEAU-LLaMA - 10%** | **63.1** | **54.1** | 73.8 | **79.2** | 43.8 | **26.5** | 27.8 | 46.0 | **77.0** | **94.8** |
| Perplexity Guidance - 30% | 62.1 | 47.8 | 69.0 | 63.7 | 46.0 | 28.0 | 28.4 | 49.0 | 72.5 | 82.2 |
| CaR - 30% | 60.0 | 48.6 | 79.3 | 77.0 | 38.5 | 21.0 | 19.8 | 44.0 | 71.9 | 83.0 |
| Cherry Selection - 30% | 61.6 | 47.0 | 68.2 | 71.5 | 39.8 | 22.0 | 26.3 | 50.8 | 69.3 | 88.4 |
| **GATEAU-LLaMA - 30%** | **64.1** | **50.4** | 78.0 | 73.5 | 42.0 | 24.5 | **29.5** | 46.8 | **73.8** | **92.1** |
| Perplexity Guidance - 50% | 62.3 | 49.6 | 79.0 | 71.0 | 47.3 | 24.5 | 28.0 | 42.0 | 69.5 | 86.3 |
| CaR - 50% | 61.6 | 47.9 | 74.0 | 77.3 | 39.0 | 21.5 | 24.5 | 42.0 | 67.8 | 91.8 |
| Cherry Selection - 50% | 62.9 | 49.6 | 77.8 | 76.2 | 48.3 | 22.5 | 30.5 | 35.8 | 68.2 | 91.5 |
| **GATEAU-LLaMA - 50%** | **64.1** | **50.4** | 78.0 | 73.5 | 42.0 | **24.5** | 29.5 | 46.8 | 73.8 | 92.1 |
| **Limited Short Instruction Data Settings** | | | | | | | | | | |
| w/o SFT | 43.5 | 25.6 | 44.5 | 44.0 | 35.0 | 16.5 | 18.0 | 28.0 | 42.0 | 48.8 |
| w/o Long SFT | 56.4 | 44.5 | 66.3 | 65.8 | 46.5 | 21.0 | 23.5 | 38.3 | 63.5 | 79.1 |
| Full - 100% | 54.5 | 40.9 | 65.8 | 56.0 | 35.5 | 21.0 | 23.5 | 34.0 | 67.5 | 78.3 |
| Perplexity Guidance - 10% | 61.9 | 39.5 | 73.8 | 61.8 | 39.3 | 27.5 | 29.1 | 47.1 | 58.5 | 72.3 |
| CaR - 10% | 59.3 | 40.3 | 66.5 | 64.3 | 49.3 | 21.5 | 26.3 | 28.8 | 62.0 | 80.5 |
| Cherry Selection - 10% | 53.0 | 42.3 | 56.8 | 72.3 | 39.5 | 17.0 | 26.5 | 34.8 | 59.3 | 75.3 |
| **GATEAU-LLaMA - 10%** | **62.2** | **44.6** | 69.9 | 67.5 | 39.8 | 24.0 | 27.5 | **50.7** | **66.3** | **83.0** |
| Perplexity Guidance - 30% | 58.9 | 41.4 | 69.4 | 68.0 | 37.0 | 28.5 | 28.9 | 47.8 | 57.8 | 64.8 |
| CaR - 30% | 52.8 | 44.3 | 67.0 | 66.5 | 37.3 | 25.0 | 24.8 | 28.5 | 68.5 | 71.0 |
| Cherry Selection - 30% | 54.8 | 36.6 | 67.5 | 57.5 | 34.0 | 19.5 | 20.4 | 35.5 | 63.5 | 69.7 |
| **GATEAU-LLaMA - 30%** | **62.0** | **43.7** | 62.0 | 65.7 | **45.4** | 27.5 | **31.7** | 41.7 | **71.7** | 72.0 |
| Perplexity Guidance - 50% | 57.6 | 40.9 | 59.5 | 74.5 | 41.0 | 25.0 | 26.0 | 37.3 | 55.3 | 75.3 |
| CaR - 50% | 58.3 | 44.1 | 70.0 | 67.2 | 43.3 | 25.5 | 30.5 | 28.5 | 71.5 | 73.5 |
| Cherry Selection - 50% | 57.7 | 41.4 | 70.0 | 63.2 | 37.5 | 18.3 | 26.3 | 43.9 | 61.1 | 76.5 |
| **GATEAU-LLaMA - 50%** | **64.2** | **44.1** | 61.5 | 67.0 | **46.3** | **28.0** | **31.4** | **47.0** | 65.8 | **84.3** |

mode as recommended by MT-Bench's authors. This mode asks GPT-4 to grade and give a score to the model's answer directly without pairwise comparison. For each turn, GPT-4 will give a score on a scale of 10. We then compute the average score on all turns.

**More Detailed Results.** We show the detailed results of MT-Bench in Table 7.

### C.4 NEEDLE IN THE HAYSTACK TEST

For the "Needle in A Haystack" evaluation, following the same original configuration as the original method (Gkamradt, 2023), we use "The best thing to do in San Francisco is eat a sandwich and sit in Dolores Park on a sunny day." as the needle fact, and Paul Graham's essays as the long haystack context. We use the same prompt as Bai et al. (2024): "What is the best thing to do in San Francisco? Here is the most relevant sentence in the context:".

### C.5 GPT-4 VERSION

For all the evaluations using the GPT-4 (evaluations for LongBench-Chat, LongBench, MT-Bench, and Needle in the Haystack test), we used `GPT-4 API` in August 2024. It ensures that we keep the same as Bai et al. (2024). According to the documents from OpenAI [1], `GPT-4 API` currently points to `GPT-4-0613 API`.

## D FURTHER EXPLORATION

### D.1 GENERAL CHARACTERISTICS OF SELECTED SAMPLES FROM GATEAU

Utilizing GPT-4, we evaluate each sample on five aspects: the coherence of long input contexts, the necessity of long input contexts, helpfulness of response, the faithfulness of response, and the complexity of instruction. Different from the previous GPT-4 evaluation detailed in the Appendix C.5, we use `GPT-4-Turbo API` (now points to `GPT-4-Turbo-2024-04-09`) as our evaluator, as this version of API has larger context window to conduct the more correct evaluation for our long input contexts. The prompt for GPT-4 evaluation on the coherence of long input contexts is:

---

[1] https://platform.openai.com/docs/models/gpt-4-turbo-and-gpt-4

> **Evaluation Prompt for the Coherence of Long Input Contexts**
>
> You are asked to evaluate the Long Input Contexts as an impartial judge, and your evaluation should follow these scoring principles:
> 1. Read the given Long Input Contexts carefully.
> 2. Evaluate the fluency and coherence of Long Input Contexts.
> 3. Evaluate whether the Long Input Contexts are focused and relevant.
>
> Please give an overall integer rating from 1 to 5 based on the above principles, strictly in the following format:"[[rating]]", e.g. "[[5]]".
>
> Please rate the following Long Input Contexts based on the scoring principles:
>
> [Long Input Contexts begins]
> {*Long Input Contexts*}
> [Long Input Contexts ends]
>
> Rating:

The prompt for GPT-4 evaluation on the necessity of long input contexts is:

> **Evaluation Prompt for the Necessity of Long Input Contexts**
>
> You are asked to evaluate the Long Input Contexts as an impartial judge, and your evaluation should follow these scoring principles:
> 1. Read the given Instruction, Long Input Contexts and Assistant's answer carefully.
> 2. Evaluate how difficult to get Assistant's following the given Instruction without the given Long Input Contexts.
> 3. Evaluate how necessary the given Long Input Contexts are to get the Assistant's answer. If the Long Input Contexts is meaningless or irrelevant, points should be deducted from the overall score.
>
> Please give an overall integer rating from 1 to 5 based on the above principles, strictly in the following format:"[[rating]]", e.g. "[[5]]".
>
> [Instruction begins]
> {*Instruction*}
> [Instruction ends]
>
> [Long Input Contexts begins]
> {*Long Input Contexts*}
> [Long Input Contexts ends]
>
> Please rate the following assistant answers based on the scoring principles:
>
> [Assistant's answer begins]
> {*Assistant's answer*}
> [Assistant's answer ends]
>
> Rating:

The prompt for GPT-4 evaluation on the faithfulness of response is:

> **Evaluation Prompt for the Faithfulness of Response**
>
> You are asked to evaluate the AI assistant's answers to user questions as an impartial judge, and your evaluation should follow these scoring principles:
> 1. Read the given Instruction, Long Input Contexts and Assistant's answer carefully.
> 2. Identify all errors in the AI Assistant's answers and consider how much they affect the answer to the question.

3. Evaluate how faithful the AI assistant's answers are to follow the Instruction, i.e., how correct and closely related to the Instruction.
4. Evaluate how faithful the AI assistant's answers are based on the Long Input Contexts, i.e., how correct and closely related to the Long Input Contexts.

Please give an overall integer rating from 1 to 5 based on the above principles, strictly in the following format:"[[rating]]", e.g. "[[5]]".

[Instruction begins]
{*Instruction*}
[Instruction ends]

[Long Input Contexts begins]
{*Long Input Contexts*}
[Long Input Contexts ends]

Please rate the following assistant answers based on the scoring principles:

[Assistant's answer begins]
{*Assistant's answer*}
[Assistant's answer ends]

Rating:

The prompt for GPT-4 evaluation on the helpfulness of response is:

### Evaluation Prompt for the Helpfulness of Response

You are asked to evaluate the AI assistant's answers to user questions as an impartial judge, and your evaluation should follow these scoring principles:
1. Read the given Instruction and Assistant's answer carefully.
2. Identify all errors in the AI Assistant's answers and consider how much they affect the answer to the question.
3. Evaluate how helpful the AI assistant's answers are in directly answering the user's questions and providing the information the user needs.

Please give an overall integer rating from 1 to 5 based on the above principles, strictly in the following format:"[[rating]]", e.g. "[[5]]".

[Instruction begins]
{*Instruction*}
[Instruction ends]

Please rate the following assistant answers based on the scoring principles:

[Assistant's answer begins]
{*Assistant's answer*}
[Assistant's answer ends]

Rating:

The prompt for GPT-4 evaluation on the complexity of instruction is:

### Evaluation Prompt for the Complexity of Instruction

You are asked to evaluate the Instruction as an impartial judge, and your evaluation should follow these scoring principles:
1. Read the given Instruction carefully.
2. Evaluate the scope of the Instruction, i.e., whether the Instruction encompasses information necessary for successful completion.

Table 8: Further exploration of Homologous Model's Guidance.

| Model | LongBench | | | LongBench-Chat | MT-Bench | | |
|---|---|---|---|---|---|---|---|
| | Single-Doc QA | Multi-Doc QA | Summarization | Avg | First-turn | Second-turn | Avg |
| **Real-world Settings** | | | | | | | |
| **GATEAU-LLaMA - 50%** | **38.9** | **25.8** | 25.5 | **56.8** | **64.1** | 50.4 | **57.3** |
| -w/o Extended Context Windows | 38.1 | 25.4 | **25.6** | 55.8 | 63.7 | 50.6 | 57.1 |
| -w/o Norm in Eq. (2) | 37.5 | 24.1 | 25.3 | 56.2 | 64.1 | 50.4 | 57.3 |
| Homologous Model's Guidance | 38.4 | 24.3 | 25.1 | 53.2 | 61.7 | 51.5 | 56.6 |
| Perplexity Guidance | 37.9 | 23.4 | 25.4 | 49.8 | 62.3 | 49.6 | 55.9 |
| Non-Homologous Model's Guidance | 37.2 | 23.2 | 24.8 | 48.2 | 59.2 | 49.3 | 54.3 |
| **Limited Short Instruction Data Settings** | | | | | | | |
| **GATEAU-LLaMA - 50%** | **31.0** | 18.1 | **25.3** | **59.0** | **64.2** | **44.1** | **54.2** |
| -w/o Extended Context Windows | 29.2 | **18.8** | 25.2 | 57.6 | 60.2 | 44.0 | 52.1 |
| -w/o Norm in Eq. (2) | 29.7 | 18.7 | 24.9 | 55.2 | 62.0 | 40.1 | 51.1 |
| Homologous Model's Guidance | 28.5 | 17.5 | 24.7 | 53.2 | 61.3 | 42.4 | 51.8 |
| Perplexity Guidance | 28.3 | 16.8 | 24.7 | 51.0 | 57.6 | 40.9 | 49.2 |
| Non-Homologous Model's Guidance | 28.7 | 16.8 | 24.8 | 50.2 | 60.1 | 40.3 | 50.2 |

3. Evaluate the depth of the Instruction, i.e., whether the Instruction provides thorough details and nuances.

4. Evaluate whether Instruction integrates multiple steps or concepts that require careful attention and understanding.

5. If the Instruction is too easy to follow, points should be deducted from the overall score.

Please give an overall integer rating from 1 to 5 based on the above principles, strictly in the following format:"[[rating]]", e.g. "[[5]]".

Please rate the following Instruction based on the scoring principles and examples above:

[Instruction begins]
{*Instruction*}
[Instruction ends]

Rating:

## D.2 Further Exploration of Homologous model's guidance

We further explore some key questions in the Homologous Model's Guidance.

**Why Do We Need Homologous Models?** Homologous Model's Guidance (HMG) aims to assess the degree of long-range dependencies required for the corresponding response generation, by comparing the perplexity scores of the response between two homologous models with different context windows. The idea behind HMG is that the primary difference between homologous models with varying context windows lies in their different capabilities for modeling long-range dependencies instead of other capabilities. Thus, the disparity in the perplexity scores can be interpreted as reflecting the difference in the long-range dependencies modeling capabilities required to generate the given response. To evaluate the effectiveness of our idea, we replace *LLaMA-2-7B-base-4k* with *Qwen-2-7b-base-8k* (Yang et al., 2024) as model $\theta_A$ in Eq. (2), namely *Non-Homologous Model's Guidance*. As shown in Table 8, we find *Non-Homologous Model's Guidance* achieve worse performance than *Homologous Model's Guidance* in two designed settings. It shows that HMG can exclusively measure the richness of long-range dependency relations in long SFT samples. As non-homologous models have different pre-training phases and model architectures, the modified Eq. (2) can not effectively measure the degree of long-range dependencies required for response generation and introduce the influence brought by other different capabilities of non-homologous models, resulting in the worse performance.

**Why Do We Apply Normalization in Eq. (2) ?** We apply softmax normalization to each score in Eq. (2) to determine its respective ranking among the datasets for two perplexity scores. This is because our early experiments observed that applying softmax normalization can further improve the performance shown in Table 8. This may due to the fact that some extremely noisy samples tend to have large perplexity scores, which in turn lead to unstable HMP scores if we do not apply normalization in Eq. (2). Training LLMs on these noisy samples further leads to poor results.

**What Will Happen if We Do Dot Extend the Context Windows of LLaMA-2-4k?** Our early experiments also explore what will happen if we do not extend the context windows of model $\theta_A$

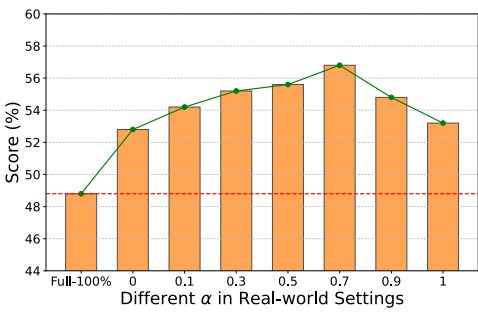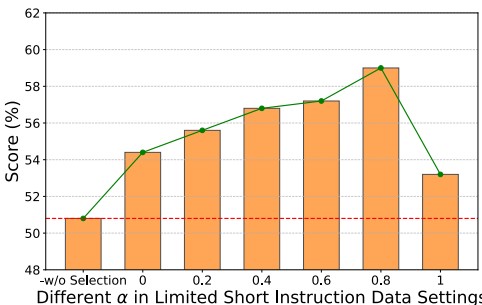

Figure 6: Results (%) on LongBench-Chat with different hyperparameter $\alpha$ in Eq. (6).

in Eq. (2). As shown in Table 8, we are surprised to find that *-w/o Extended Context Windows* also achieves competitive results in three benchmarks compared to *GATEAU-LLaMA*. Even the perplexity score $PPL_{\theta_A}(y|c, x)$ from the model $\theta_A$ can be very large, e.g., the value of $PPL_{\theta_A}(y|c, x)$ can be larger than 1000, the value after softmax normalization is still useful and applicable in the Homologous Models' Guidance. This interesting finding can be used to reduce the complexity of applying Homologous Models' Guidance and achieve competitive performance.

### D.3 PARAMETER STUDY

As shown in Figure 6, we conduct experiments to explore the impact of important hyperparameter $\alpha$ in Eq. (6) to further understand our method. We report the results of *GATEAU-LLaMA-50%* on LongBench-Chat in two settings. Overall, although the choice of different $\alpha$ will have some impact on the LLM's performance, the performance will always be improved over the baseline *Full-100%*, i.e., using the whole training dataset without data selection. Meanwhile, we also find that using both the Homologous Model's Guidance and Contextual Awareness Measurement will further improve the performance than only using one of them. This is because the Homologous Model's Guidance and Contextual Awareness Measurement attempt to measure the difficulty brought by the long-range dependencies from two different perspectives, i.e., separately measuring the difficulty of generating corresponding responses and understanding long input contexts due to the long-range dependencies.

### D.4 HUMAN EVALUATION

During the human evaluation, the participants follow these principles to make the decision:

> **The Principles of Human Evaluation**
>
> You are asked to evaluate the responses generated by different models, and your evaluation should follow these principles:
> 1. **Correctness**: Focus primarily on whether the response covers the key points in the reference answer. For reference answers containing multiple key points, look for how many of these the response accurately addresses and score accordingly.
> 2. **Helpfulness**: Evaluate how helpful the AI assistant's answers are in directly answering the user's questions and providing the information the user needs.
> 3. **Faithfulness**: Examine any additional information in the AI assistant's answer to ensure that it is correct and closely related to the question. If this information is incorrect or not relevant to the question, points should be deducted from the overall score.
> 4. **Coherence**: Evaluate how fluent and coherent the AI assistant's answers are. Also, consider deducting points for overly verbose responses or those that are excessively generalized.
>
> Finally, please make a decision among 3 opinions, including Win, Tie, and Loss.

If the final result can not be determined by the majority voting of three participants, we will make the discussion among the participants and vote on the result again.

### D.5 OOD Case Study

As part of our research on aligning LLMs on long context, we further come up with a practical case study. We consider using an out-of-distribution (OOD) query, that is, the query that the model has not encountered in the long context SFT data. Specifically, we select the *Biden-Trump debate transcript* [2] from the 2024 election season as the OOD query, because this debate is organized subsequent to the collection of our used training datasets.

---

**OOD Case Study**

**👤 User:**

[Debate transcript begins]
{ }
[Debate transcript ends]

[Instruction begins]
As a professional summarizer, create a concise and comprehensive summary of the provided debate, while adhering to these guidelines:
1. Craft a summary that is detailed, thorough, in-depth, and complex, while maintaining clarity and conciseness.
2. Use proper grammar, punctuation, and spelling.
3. Write in a formal, objective, and impartial tone.
4. Provide a clear and accurate summary of the debate, without bias or personal opinions.
[Instruction ends]

Summary:

**😀 Ours:**

The CNN presidential debate between President Joe Biden and former President Donald Trump was held in Atlanta, Georgia. The debate was moderated by CNN anchors Jake Tapper and Dana Bash. The debate covered various topics including the economy, immigration, foreign policy, democracy, and more. On the economy, President Biden defended his administration's actions, stating that he inherited a disastrous situation from the previous administration. He highlighted his administration's job creation efforts and infrastructure spending. Former President Trump criticized Biden's policies, claiming that Biden's tax hikes and regulations were hurting the economy. On immigration, President Biden defended his administration's handling of the border crisis, stating that he had increased border patrol and asylum officers. Former President Trump criticized Biden's immigration policies, claiming that Biden was allowing "terrorists" to enter the country. On foreign policy, President Biden criticized former President Trump's handling of foreign affairs, particularly in regards to Russia and Ukraine. He stated that Trump had encouraged Putin to invade Ukraine and that Trump had not done enough to stop the war. Former President Trump defended his actions, stating that he had put pressure on Ukraine to investigate Biden's son and that he had not done enough to stop the war because he was focused on China. On democracy, President Biden criticized former President Trump's handling of the January 6th attack on the Capitol, stating that Trump had encouraged the protesters to go to the Capitol. Former President Trump defended his actions, stating that he had spoken peacefully and patriotically. Overall, the debate was contentious and both candidates strongly defended their records and policies.

---

[2]https://edition.cnn.com/2024/06/27/politics/read-biden-trump-debate-rush-transcript/index.html

☹ **Full - 100%:**

The debate between President Joe Biden and former President Donald Trump was held in Atlanta, Georgia, on October 22, 2020. The debate was moderated by CNN anchors Jake Tapper and Dana Bash. The debate focused on various issues such as the economy, immigration, foreign policy, democracy, and more. On the issue of the economy, President Biden argued that he inherited a disastrous economy from Trump, with high unemployment rates and no jobs. He claimed that his policies have helped create jobs and increase wages for working-class Americans. Trump, on the other hand, claimed that he had the best economy in history and that Biden's policies have caused inflation and driven up prices. On the issue of immigration, Trump claimed that Biden was allowing "millions of people to come in and destroy our country." He also claimed that Biden was responsible for the worst border crisis in history. Biden argued that Trump had left the border in chaos and that his policies had caused the crisis. On foreign policy, Trump claimed that he had brought the United States to the brink of war with Iran and that he had made a deal with North Korea that was a "disaster." Biden argued that Trump had put the United States in a dangerous position with his rhetoric and that he had made the world a more dangerous place. On democracy, Trump claimed that the election was rigged and that he had won the election. Biden argued that Trump had incited a violent insurrection at the Capitol and that he had undermined democracy. In conclusion, the debate was contentious and both candidates presented their respective arguments and policies. However, it remains to be seen how the voters will interpret the debate and how it will impact the election outcome.

We show the results generated by *GATEAU-LLaMA-30%* and *Full-100%* in Real-world settings. We highlight the sentences that can be easily misunderstood or contain factual errors (e.g., this debate is organized in 2024 instead of 2020). We can find that our method achieves better faithfulness and fluency, showing the effectiveness of our method in handling OOD queries.

