# OpenReview forum: "Selecting Influential Samples for Long Context Alignment via Homologous Models’ Guidance and Contextual Awareness Measurement"
_ICLR.cc/2025/Conference — ICLR 2025 Conference Withdrawn Submission_

### Official Review · Reviewer_93rU · 2024-11-02

**Soundness:** 2
**Presentation:** 3
**Contribution:** 3
**Rating:** 5
**Confidence:** 4

**Summary:**

This paper introduces GATEAU, a data selection framework designed to identify critical instruction samples for long context alignment of LLMs. Specifically, GATEAU consists of two core modules: Homologous Models' Guidance (HMG) and Contextual Awareness Measurement (CAM). HMG assesses the difficulty of generating responses due to long-range dependencies by comparing the perplexity scores of homologous models with varying context windows. CAM measures how effectively a model focuses on relevant segments of extended input. The outputs of HMG and CAM are combined to produce a final ranking criterion for data selection. Extensive experiments indicate that the proposed method significantly enhances the long-context instruction-following capabilities of LLMs.

**Strengths:**

1. This paper is well motivated. Identifying high-quality long instruction-following samples is essential for improving long context alignment.
2. The proposed GATEAU aligns naturally with the challenges of long-range dependency modeling, making the methodology intuitive and easy to understand.
3. The authors demonstrate the effectiveness of GATEAU through extensive experiments across multiple benchmarks. The results consistently show that the selected samples significantly improve model performance on both long and short instruction-following tasks.

**Weaknesses:**

1. The proposed method may assign similar high scores to duplicate or highly similar samples, assuming they contribute independently to model improvement. However, repeated exposure to similar samples may not add incremental value and could even undermine the alignment process of LLMs.
2. The paper does not discuss the computational resources required to implement the GATEAU framework. A selection method that is slow or computationally prohibitive is less feasible in practical scenarios, especially for extremely long instruction-following examples.
3. Although the paper includes ablation studies, it lacks an experiment testing the effectiveness of using a single model's perplexity (e.g., LLaMA-2-7B-base-64k) as the sole criterion for data selection. Adding this experiment would provide valuable insight into the authors' claim on page 4, line 174, that high perplexity alone does not adequately reflect response difficulty in long-context scenarios.

**Questions:**

1. How does the proposed framework handle potentially redundant or highly similar samples? Could repeated high-scoring samples lead to an overrepresentation of certain types of data, thus limiting the diversity and richness of long-context dependencies in the selected dataset?
2. Would this approach remain feasible for datasets of larger scale, and what are the expected costs in terms of time and resources?
3. Is there some display problem in Figure 2?

---

> ### Author Response · Authors · 2024-11-22
> **Response to Reviewer 93rU (1/N)**
>
> Thanks for your valuable review and suggestions! It is encouraging to see you find our methodology well-motivated and effective.
>
> We sincerely thank you for your time and constructive comments. Below, we provide detailed replies to your comments to resolve your concerns.
>
> ###  **W1 and Q1:**
>
> > **W1:** The proposed method may assign similar high scores to duplicate or highly similar samples, assuming they contribute independently to model improvement. However, repeated exposure to similar samples may not add incremental value and could even undermine the alignment process of LLMs.
> >
> >
> > **Q1:** How does the proposed framework handle potentially redundant or highly similar samples? Could repeated high-scoring samples lead to an overrepresentation of certain types of data, thus limiting the diversity and richness of long-context dependencies in the selected dataset?
> >
>
> ###  **Response:**
> This is an interesting question. In our long SFT data selection process, we have partially considered such sample redundancy. In particular, HMG and CAM separately measure the difficulty of generating corresponding responses and understanding long input contexts due to the long-range dependencies, thus the final score derived from two different perspectives inherently reduces the influence of redundant samples. **As shown in Table 7 in the Appendix, our method achieves better overall performance and more balanced performance in 8 different tasks, showing the effectiveness and diversity of selected samples by GATEAU.**
>
> Meanwhile, to further explore whether our method inherently reduces the influence of redundant samples, we employ the cluster model as [1] to cluster all candidate instruction pairs into k clusters. Specifically, we employ the k-Means algorithm and a sentence transformers model which is used to map sentences to a 384-dimensional dense vector space. Subsequently, semantic features are PCA-reduced to retain 95% of dimensions. Finally, by setting the number of clusters as $k =  \sqrt{n/2}$ for $n$ long SFT samples, all 10k long SFT samples are clustered into 70 clusters. Finally, all samples are sorted based on their scores according to Eq. (6), and the top $n_1$ samples are selected. Within each cluster, samples are sorted by score, and the top $n_2$ pairs are chosen. We set $n_2$ to 1, which is the same as [1]. Finally, we can get $n_1 + k * n_2$ (i.e., $4300 + 70 * 1$) samples and use these selected data to train the model, namely -w Diversity-preserved Selection. We report the results of GATEAU-LLaMA - 50\% on LongBench-Chat and MT-Bench in two settings.
>
> | Method | **LongBench-Chat** | **MT-Bench** |
> | --- | --- | --- |
> | GATEAU-LLaMA - 50% in Real-world Settings | **56.8** | 57.3 |
> | -w Diversity-preserved Selection | 56.2 | **57.8** |
> | GATEAU-LLaMA - 50% in Limited Short Instruction Data Settings | 59.0 | **54.2** |
> | -w Diversity-preserved Selection | **59.2** | 53.4 |
>
> **In this table, we find that using the Diversity-preserved Selection does not consistently improve the final performance, showing our proposed GATEAU has partially addressed the sample redundancy and implicitly ensured the diversity of selected long SFT data.**
>
> [1]  Clustering and ranking: Diversity-preserved instruction selection through expert-aligned quality estimation. EMNLP 2024

---

> ### Author Response · Authors · 2024-11-22
> **Response to Reviewer 93rU (2/N)**
>
> ###  **W2 and Q2:**
> >  **W2:** The paper does not discuss the computational resources required to implement the GATEAU framework. A selection method that is slow or computationally prohibitive is less feasible in practical scenarios, especially for extremely long instruction-following examples.
> >
> > **Q2:** Would this approach remain feasible for datasets of larger scale, and what are the expected costs in terms of time and resources?
> ### **Response:**
>
> Thanks for your advice, we believe adding such details will help our paper be more comprehensive.
>
> <1> **Experimental Device**: As shown in the Appendix A, experiments are conducted on 8xA800 80G GPUs (experiments are not limited to this type of GPU).
>
> <2> **GPU Execution Time:** Based on the principle of making full use of GPUs (e.g.,  multi-processing and selecting large batch sizes, etc), we list the execution time in following table:
>
> | **Stage** | **Execution Time** |
> | --- | --- |
> | Training on the full dataset in the real-world setting | ~176 GPU hours |
> | Selecting long SFT data via HMG | ~64 GPU hours |
> | Selecting long SFT data via CAM | ~48 GPU hours |
> | Selecting long SFT data via Cherry Selection | ~80 GPU hours |
> | Selecting long SFT data via Perplexity Guidance | ~32 GPU hours |
>
> As shown in this table, we can find our method (HMG + CAM) introduces acceptable offline time overhead compared to the supervised fine-tuning stage and improves the overall performance of long-context LLMs. Perplexity Guidance applies a single LLM to compute the score, thus it achieves less execution time but worse performance in our experiments. Meanwhile, another strong baseline Cherry Selection introduces an additional training stage and computes the proposed Instruction-Following Difficulty (IFD) by applying the forward propagation twice on a single long SFT data, thus necessitating more execution time compared to our proposed HMG. Meanwhile, our CAM and HMG can process the data in parallel to further decrease the execution time, e.g., only 8 hours with 16xA800 80G GPUs.  **Overall, compared to other baselines, the experimental results of our proposed GATEAU (consists of HMG and CAM) demonstrate that the additional execution time is worthwhile.**
>
> <3> **GPU Memory Burden:** As our method is designed to score the long SFT data, and then select the influential samples used for the SFT stage, thus our method does not introduce additional memory burden during the supervised fine-tuning and inference stage of the long-context model $\theta_{B}$. For your concerns about HMG, we compute perplexity scores generated from two models $\theta_{A}$ and $\theta_{B}$ for a given SFT data in parallel, and use the computed perplexity scores  (cached in JSON files) to get the homologous models’ perplexity score HMP as shown in Eq. (2). **Thus HMG does not introduce additional GPU memory burden, only introducing acceptable additional execution time as shown in Execution Time Table**. The GPU memory requirements of CAM tem from the calculation of the attention scores for lengthy inputs, as well as the perplexity score computation. **This process is equivalent to performing two forward passes over the dataset without updating gradients, thus it does not add an extra GPU memory burden.**
>
> \\\\
>
> ###  **W3:**
> > **W3:** Although the paper includes ablation studies, it lacks an experiment testing the effectiveness of using a single model's perplexity (e.g., LLaMA-2-7B-base-64k) as the sole criterion for data selection. Adding this experiment would provide valuable insight into the authors' claim on page 4, line 174, that high perplexity alone does not adequately reflect response difficulty in long-context scenarios.
> ### **Response:**
> Thank you for your suggestion! **Actually, we have already compared this important baseline, namely Perplexity Guidance in our paper, in our comprehensive experiments (e.g., Table 1, Table 2, Table 3, Table 4, and Table 8 in the Appendix).** We find that while using a single model's perplexity can be used as weak but positive guidance to select SFT samples compared to using the full long SFT dataset, our proposed GATEAU achieved consistently better performance, showing that high perplexity alone does not adequately reflect response difficulty in long-context scenarios.
>
> \\\\
>
> ### **Q4:**
> > **Q4:** Is there some display problem in Figure 2?
> ### **Response:**
> Figure 2 illustrates the "Needle in the Haystack" test, which evaluates the model’s ability to utilize information from 10 different positions within long contexts of varying lengths (1k–60k). The results are displayed using different colors. The green color across Figure 2 indicates that the model with GATEAU successfully passes all settings, resulting in a **uniform green display**. This comparison highlights the effectiveness of our method.
>
> \\\\
>
> We hope this detailed explanation successfully addresses your concern. Your advice has significantly contributed to the quality of our paper.

---

> > ### Comment · Reviewer_93rU · 2024-11-26
> > **Acknowledgement**
> >
> > This message acknowledges the authors' response. Regarding the first question, it seems the repetition has a mixed impact on the SFT performance, which may have significant impacts. Besides, this question also raise the concern on performance degradation on other general LLM capabilities. It is suggested to have further investigation into this interesting pattern. Regarding the second question, it seems the authors didn't directly answer the question on large-scale data. Based on current response, I choose to maintain current score.

---

> > > ### Author Response · Authors · 2024-11-26
> > >
> > > Dear Reviewer 93rU,
> > >
> > >
> > >
> > > Thank you for your valuable feedback. We would like to provide some additional explanations and details to clarify any misunderstandings between us.
> > >
> > >
> > >
> > > **<1> There were no data repetition issues in our experiments:** We use LongAlign[1] as the long SFT dataset in our experiments, which ensures that each data point is unique, **thus the data repetition issue you mentioned does not occur in our experiments**. Additionally, we have considered the similarity of data, i.e., the diversity of the data selected by GATEAU, details of which can be seen in <2>.
> > >
> > >
> > >
> > > **<2> Our method does not limit the diversity of selected data:** In our paper, we compare our GATEAU with the state-of-the-art method that considers data diversity, named CaR[2], and find that our method GATEAU consistently outperforms CaR. Meanwhile, as shown in Table 7 in the Appendix, our method achieves **better overall performance and more balanced performance in 8 different tasks**, showing the effectiveness and diversity of selected samples by GATEAU. In the additional experiments detailed in our response to you, we also find that integrating a module that focuses on preserving data diversity (namely -w Diversity-preserved Selection) into our existing method GATEAU did not consistently improve the final performance, **suggesting that GATEAU does not limit the diversity of the selected data.**
> > >
> > >
> > >
> > > **<3> Our method is applicable on large-scale long SFT datasets:** In our previous response, we provide detailed data demonstrating that our method does not introduce any additional GPU burdens, and only incurs an acceptable increase in execution time, making it easily applicable to large-scale long SFT dataset. **Additionally, the long SFT dataset LongAlign[1] that we used in our experiments is one of the largest available Long SFT datasets, containing 10,000 long SFT data.**
> > >
> > >
> > >
> > > We hope our response has addressed your concerns, and we look forward to your further feedback.
> > >
> > >
> > > [1] LongAlign: A Recipe for Long Context Alignment of Large Language Models. EMNLP 2024 Findings
> > >
> > > [2] Clustering and ranking: Diversity-preserved instruction selection through expert-aligned quality estimation. EMNLP 2024

---

> ### Author Response · Authors · 2024-11-26
>
> Dear Reviewer 93rU,
>
> We would like to thank you again for your detailed reviews. We have updated our draft and added replies to your concerns with our latest experimental results.
>
> Since the rebuttal deadline is approaching soon. Given that your current score is 5, we would appreciate it if you could let us know if our responses have addressed your concerns satisfactorily. If your concerns have not been resolved, could you please let us know about it so that we have the opportunity to respond before the deadline?
>
> We would be happy to have any follow-up discussions or address any additional concerns.
>
> Thanks very much! Looking forward to your reply.
>
> Best,
>
> Authors

---

> ### Author Response · Authors · 2024-12-04
>
> Dear Reviewer,
>
> Since it is the last day of the discussion, we hope that you can take a look at our response. Thanks.
>
> Best regards,
>
> The Authors

---

### Official Review · Reviewer_KGEd · 2024-11-02

**Soundness:** 3
**Presentation:** 3
**Contribution:** 2
**Rating:** 5
**Confidence:** 3

**Summary:**

The paper introduces GATEAU, a framework aimed at improving the alignment of large language models with long-context data by identifying influential samples. GATEAU leverages two main components: Homologous Models’ Guidance, which assesses the difficulty of generating responses based on perplexity differences between models with varying context windows, and Contextual Awareness Measurement, which evaluates whether models focus on crucial segments in long input contexts. Through these mechanisms, GATEAU selectively curates samples to enhance LLMs' ability to follow instructions with long-range dependencies.

**Strengths:**

- The paper addresses an important challenge of optimizing data selection for long-context alignment, enhancing LLMs' performance in real-world applications that require handling long, complex contexts.
- The paper provides comprehensive evaluations across multiple tasks and varied compression ratios, which helps illustrate the model’s versatility and effectiveness.
- The methodology for Homologous Models’ Guidance is well-motivated. Leveraging LLM collaboration for data selection is interesting.
- Ablation study and human evaluation are conducted to verify the effectiveness of the method.

**Weaknesses:**

- The study only tests on the LLaMA2 family, which may restrict the generalizability of findings. Additionally, HMG relies on perplexity differences between homologous models with different context windows, leaving unclear guidance on applying the technique to other models lacking such variants.
- There is no analysis of the time efficiency of the data selection method. Since CAM requires estimating importance scores for each segment, which may be computationally expensive, time efficiency is a critical factor.
- The CAM module’s process for calculating attention weights is vague, particularly in obtaining the attention weight across tokens in response $y$ to token $t_j$. Including pseudo-code or further clarification would make the implementation more accessible.
- The method of dividing long inputs into equal segments may overlook the fact that important information could span multiple segments. Additionally, segment importance can depend conditionally on other segments, an aspect not accounted for in the current approach.
- The method averages attention across heads and layers without distinguishing them, potentially introducing noise, as different heads or layers may capture varied aspects of the input [1]. Calibrating the attention aggregation could improve focus on relevant input sections.
- Key hyperparameters, such as the number of segments, remain unexplored, missing potential for optimization in specific contexts.

[1] Model Tells You What to Discard: Adaptive KV Cache Compression for LLMs. ICLR 2024

**Questions:**

Please refer to the weakness part.

---

> ### Author Response · Authors · 2024-11-22
> **Response to Reviewer KGEd (1/N)**
>
> Thanks for your valuable review and suggestions! It is encouraging to see you find our methodology well-motivated and interesting.
>
> We sincerely thank you for your time and constructive comments. Below, we provide detailed replies to your comments to resolve your concerns.
>
> ### **W1:**
> > **W1:** The study only tests on the LLaMA2 family, which may restrict the generalizability of findings. Additionally, HMG relies on perplexity differences between homologous models with different context windows, leaving unclear guidance on applying the technique to other models lacking such variants.
> >
>
> ### **Response:**
> Thanks for your interest! Our HMG method indeed requires two homologous models with different context windows, thus limiting the range of models we can use to further conduct the experiments. However, in practical scenarios, training a powerful long-context LLM always involves homologous models with different context windows (though these models may not be open-sourced). This is because existing LLMs are often initially pre-trained on a large-scale corpus with smaller context windows due to device limitations, they then conduct continual pre-training to extend the window size. Therefore, our method still remains effective in real-world scenarios.
>
> \\\\
>
>
> ### **W2:**
> > **W2:** There is no analysis of the time efficiency of the data selection method. Since CAM requires estimating importance scores for each segment, which may be computationally expensive, time efficiency is a critical factor.
> >
>
> ### **Response:**
> Thanks for your advice, we believe adding such details will help our paper be more comprehensive.
>
> <1> **Our Experimental Device**: Firstly, we want to explain the experimental device. As shown in Appendix A, all experiments are conducted on 8xA800 80G GPUs (the experiments are not limited to this type of GPU).
>
> <2> **GPU** **Execution Time:** Based on the principle of making full use of GPU devices (e.g., using a multi-processing strategy and choosing large batch sizes, etc, we list the execution time in the following table:
>
> | **Stage** | **Execution Time** |
> | --- | --- |
> | Training on the full dataset in the real-world setting | ~176 GPU hours |
> | Selecting long SFT data via HMG | ~64 GPU hours |
> | Selecting long SFT data via CAM | ~48 GPU hours |
> | Selecting long SFT data via Cherry Selection | ~80 GPU hours |
> | Selecting long SFT data via Perplexity Guidance | ~32 GPU hours |
>
> As shown in this table, we can find our method (HMG + CAM) introduces acceptable offline time overhead compared to the supervised fine-tuning stage and improves the overall performance of long-context LLMs. Perplexity Guidance applies a single LLM to compute the score, thus it achieves less execution time but worse performance in our experiments. Meanwhile, another strong baseline Cherry Selection introduces an additional training stage and computes the proposed Instruction-Following Difficulty (IFD) by applying the forward propagation twice on a single long SFT data, thus necessitating more execution time compared to our proposed HMG. Meanwhile, our CAM and HMG can process the data in parallel to further decrease the execution time, e.g., only 8 hours with 16xA800 80G GPUs. **Overall, compared to other baselines, the experimental results of our proposed GATEAU (consists of HMG and CAM) demonstrate that the additional execution time is worthwhile.**
>
> \\\\
>
> ### **W3:**
> > **W3:** The CAM module’s process for calculating attention weights is vague, particularly in obtaining the attention weight across tokens in response y to token tj. Including pseudo-code or further clarification would make the implementation more accessible.
> >
>
> ### **Response:**
>
> Sorry for the misunderstanding, we first compute attention weights for each token in response $y$ to the token $t_j$ in the segment $s_i$. Then we average the attention weights from each token in response $y$ to get the score $Attn_{\theta}(t_{j}|y;\theta)$ in Eq. (4), then compute the $Attn_{\theta}(s_i)$ according to Eq. (4). Meanwhile, we harness the attention weights averaged across different decoder layers and attention heads to thoroughly model how the LLM utilizes the long input contexts according to [1].
>
> [1] Found in the middle: Calibrating positional attention bias improves long context utilization. ACL Findings 2024

---

> ### Author Response · Authors · 2024-11-22
> **Response to Reviewer KGEd (2/N)**
>
> ### **W4 and W6:**
>
> > **W4:** The method of dividing long inputs into equal segments may overlook the fact that important information could span multiple segments. Additionally, segment importance can depend conditionally on other segments, an aspect not accounted for in the current approach.
> >
> >
> > **W6:** Key hyperparameters, such as the number of segments, remain unexplored, missing potential for optimization in specific contexts.
> >
>
> ### **Response:**
>
> Our CAM aims to evaluate whether LLMs’ attention is appropriately focused on important segments within the long input contexts. We separately compute the designed important score for each segment to the given response and calculate the LLM’s attention weights on each segment. In this way, even if the important information could span multiple segments, our method still calculates scores for several different segments to get the final CAS. **As shown in our experiments, our method achieves better performance in Multi-doc QA and long-context Summarization, showing that our method can effectively handle the scenario that important information could span multiple segments**. Meanwhile, Therefore, we further explore the impact of the important hyperparameter segment length. Intuitively, an excessively large segment length tends to prevent the model from focusing on the fine-grained information within the segment, whereas an excessively small segment length can lead to more semantically incoherent segments. We report the results of GATEAU-LLaMA - 50\% on LongBench-Chat in Real-world Settings.
>
> | **Length of Segment** | LongBench-Chat |
> | --- | --- |
> | 64 | 55.2 |
> | **128 (reported in our paper)** | **56.8** |
> | 256 | 56.2 |
> | 512 | 56.4 |
> | 1024 | 54.4 |
> | 2048 | 53.6 |
> | Full - 100% | 48.8 |
> | -w/o CAM | 53.2 |
>
> As shown in this table, different segment lengths affect the model's performance; however, as long as a reasonable length value is chosen, **the fluctuations in model performance are not significant**. Meanwhile, the performance will always be improved over using the whole long SFT dataset (namely Full-100%) and only using the HMG method (namely -w/o CAM), showing the effectiveness of our proposed CAM.
>
> \\\\
>
> ### **W5:**
> > **W5:** The method averages attention across heads and layers without distinguishing them, potentially introducing noise, as different heads or layers may capture varied aspects of the input. Calibrating the attention aggregation could improve focus on relevant input sections.
> >
>
> ### **Response:**
>
> This is an interesting point. Inspired by previous work[2], we harness the attention weights averaged across different decoder layers and attention heads to thoroughly model how the LLM utilizes long input contexts during response generation. In this way, as shown in our experiments, our method achieves consistently better performance. Meanwhile, choosing the optimal head and layer for long context alignment may require additional validation data and computational resources.
>
> [2] Found in the middle: Calibrating positional attention bias improves long context utilization. ACL Findings 2024
>
> \\\
>
> We trust that this additional information addresses your concerns, and we welcome any further inquiries or feedback you may have. We also hope that you can kindly increase your score if our response has helped address your concerns.

---

> ### Author Response · Authors · 2024-11-26
>
> Dear Reviewer KGEd,
>
> We would like to thank you again for your detailed reviews. We have updated our draft and added replies to your concerns with our latest experimental results.
>
> Since the rebuttal deadline is approaching soon. Given that your current score is 5, we would appreciate it if you could let us know if our responses have addressed your concerns satisfactorily. If your concerns have not been resolved, could you please let us know about it so that we have the opportunity to respond before the deadline?
>
> We would be happy to have any follow-up discussions or address any additional concerns.
>
> Thanks very much! Looking forward to your reply.
>
> Best,
>
> Authors

---

> ### Author Response · Authors · 2024-12-04
>
> Dear Reviewer,
>
> Since it is the last day of the discussion, we hope that you can take a look at our response. Thanks.
>
> Best regards,
>
> The Authors

---

### Official Review · Reviewer_pvW2 · 2024-11-07

**Soundness:** 2
**Presentation:** 3
**Contribution:** 2
**Rating:** 5
**Confidence:** 4

**Summary:**

This paper proposes a new data selection algorithm, GATEAU, for training long-context instruction following models. Gven a long-context instruction following dataset, the algorithm calculates two metrics: Homologous Models' Guidance (HMG) and Contextual Awareness Measurement (CAM). In particular, HMG measures the difficulty of generating responses due to long-range dependencies by by comparing perplexity scores of responses between two homologous models with different lengths of context windows. CAM measures the difficulty of  utilizing important parts of the long-input by evaluating whether the impoartant segments are being utilized using the attention scores. The author measures the effectiveness of the proposed method on various benchmarks (e.g., LongBench, LongBench-Chat and MT-Bench).

**Strengths:**

1. The writing is clear (except some questions I have in the Questions section below) and the author explains the proposed method well with clear formulations.
2. The author uses different benchmarks (LongBench, LongBench-Chat, MT-bench, Needle-in-a-haystack) to show the effectiveness of the proposed method on (long/short)-context instruction following tasks.

**Weaknesses:**

1.
- The method proposed heavily depends on the perplexity that measures the similarity between the base model's answer and the desired answer. The model $\theta_A$ (e.g., LLaMA-2-7B-base-64k as mentioned in the paper) in the method that calculates the perplexity is a base model (model that only does completion). It does not make sense to measure the perplexity of a base model on an instruction-following dataset because the base model wasn't trained to follow instruction. In this case, the perplexity can be high and it does not accurately measures the difficulty of a document. The author also acknowledges this and in the first metric (HMG) uses a homologous short context model $\theta_B$ as a reference model and calculates the difference between PPL of $\theta_A$ and $\theta_B$. The author claims that this "mitigate the influence brought by lacking other capabilities" (e.g., instruction-following capability, long-context capability). However, there is no guarantee that lacking other capabilities contribute equally in increasing PPL of two models. Essentially the data where the models measure the PPL on is out-of-distribution data (both models were not trained on the instruction-following data and $\theta_B$, which is extended to a longer-context window using zero-shot method, was not trained on long-context data) and model's behavior is unpredictable. Plus, $\theta_A$ was obtained by continual pretraining on long-context data. While HMG assumes $\theta_A$ and $\theta_B$ are similar models, it also depend on what the continual pretraining dataset is (here the method implicitly assumes that the additional dataset is small). If the additional dataset is very large, $\theta_A$ and $\theta_B$ can perform very differently.
- Moreover, for second metric (CAM), the LLM that measures the PPL was not trained on instruction-following dataset (and therefore the behavior is unpredictable) and there is no such reference model.

2. LESS [1] is an optimizer-based method that select a subset of instruction-following dataset by estimating data influence (selecting data points that minimizes the validation loss) and this can be as one of the baselines.

Minor issues:
1. For Table 5, current bold numbers are a bit misleading. Usually bold numbers indicate the highest numbers across some category but here it seems the proposed method is bold. Also, for 13B models, it seem the w/o HMG and w/o CAM settings are not reported. Is there a particular reason of not doing so (e.g., computational constraint)? The ablation study on 7B model does show the effectiveness.

[1] Xia, M., Malladi, S., Gururangan, S., Arora, S., & Chen, D. (2024). Less: Selecting influential data for targeted instruction tuning. arXiv preprint arXiv:2402.04333.

**Questions:**

1. For perplexity guidance (line 377), which $\theta$ do you use ($\theta_A$ or $\theta_B$ -- I am assuming $\theta_B$ here)?
2. What is the context length of MT-Bench? The paper mentions that MT-Bench is for short-context instruction following. Since the proposed method, GATEAU, is designed for long-context, do you have any hypothesis on why it also improves on the short-context instruction following tasks (Table 4)?

---

> ### Author Response · Authors · 2024-11-22
> **Response to Reviewer pvW2 for Weaknesses 1 (1/ 2)**
>
> Thanks for your valuable review and constructive comments. Below, we provide detailed replies to your comments to resolve your concerns.
>
> ### **W1:**
>
> > **W1:** The method proposed heavily depends on the perplexity that measures the similarity between the base model's answer and the desired answer. The model $\theta_{A}$ (e.g., LLaMA-2-7B-base-64k as mentioned in the paper) in the method that calculates the perplexity is a base model (model that only does completion). It does not make sense to measure the perplexity of a base model on an instruction-following dataset because the base model wasn't trained to follow instruction. In this case, the perplexity can be high and it does not accurately measures the difficulty of a document. The author also acknowledges this and in the first metric (HMG) uses a homologous short context model $\theta_{B}$ as a reference model and calculates the difference between PPL of $\theta_{A}$ and $\theta_{B}$. The author claims that this "mitigate the influence brought by lacking other capabilities" (e.g., instruction-following capability, long-context capability). However, there is no guarantee that lacking other capabilities contribute equally in increasing PPL of two models. Essentially the data where the models measure the PPL on is out-of-distribution data (both models were not trained on the instruction-following data and $\theta_{B}$, which is extended to a longer-context window using zero-shot method, was not trained on long-context data) and model's behavior is unpredictable. Plus, $\theta_{A}$ was obtained by continual pretraining on long-context data. While HMG assumes $\theta_{A}$ and $\theta_{B}$ are similar models, it also depend on what the continual pretraining dataset is (here the method implicitly assumes that the additional dataset is small). If the additional dataset is very large, $\theta_{A}$ and $\theta_{B}$ can perform very differently.
> >
>
> ### **Response:**
> Thanks for your valuable questions. We want to clarify the following to make our paper sound more sound.
>
> For your concerns about HMG and CAM:
>
> ><1>  **Is the perplexity score from the base model really so high that it cannot accurately measure the difficulty?**
>
> Intuitively, since the base model performs well on conditional generation tasks (e.g., continuation), it should also be able to **generate accurate perplexity scores** on the response of instruction-following data, even though the model might not be able to produce high-quality responses correctly, because these two capabilities are not the same. Therefore, we explore whether our long-text LLM $\theta_{B}$ would produce incorrect perplexity values. We calculate the average perplexity value generated by the long-text LLM $\theta_{B}$ for the entire long SFT dataset during the whole HMG process, which is **2.61**. Therefore, there is no issue of the perplexity from the base model being too high to accurately measure the difficulty.
>
> ><2> **Can the perplexity score generated from the base model be used as guidance to select influential samples?**
>
> The perplexity (PPL) of the responses computed with the base model is an intuitive metric, as it measures the difficulty of the data sample during the generation. In our experiments, we find simply using high perplexity (namely Perplexity Guidance in our paper) can also improve the performance compared with using the whole long SFT dataset, indicating that the effectiveness of the perplexity score from the base model in selecting long SFT samples. Previous work [1] also finds using the Instruction-Following Difficulty (a variant of perplexity score) computed with the base model also works in selecting SFT samples. **According to these experiments, we believe that the perplexity generated from a base model can be used as positive guidance to select SFT samples.** Therefore, the use of the perplexity score generated from the base model in our method makes sense when selecting long SFT data. Meanwhile, our method HMG is designed to minimize the influence of other factors (e.g., the limited instruction-following ability of a base model) and model the difficulty in modeling the long-range dependencies to construct the more effective guidance of long SFT data selection, and further improve overall performance. For CAM, utilizing perplexity scores to compute importance scores is also reasonable, and the experiments show improvement even when only using CAM.
>
> [1] From Quantity to Quality: Boosting LLM Performance with Self-Guided Data Selection for Instruction Tuning. NAACL 2024

---

> > ### Author Response · Authors · 2024-11-22
> > **Continue Response to Reviewer pvW2 for Weaknesses 1 (1/ 2)**
> >
> > ### **Response:**
> >
> > > **<3> Additional experiments to analyze the perplexity score generated from the base model.**
> >
> > We further conduct additional experiments to explore the effect of perplexity scores generated from the base model. In HMG, we use in-context learning technology to align the base model and use the perplexity score from the aligned model to select long SFT data. Specifically, we use the same 3 demonstration examples as URIAL [2]. In this way, we can get models more aligned without updating the parameters.
> >
> > | Method | **LongBench-Chat** | **MT-Bench** |
> > | --- | --- | --- |
> > | GATEAU-LLaMA - 50% in Real-world Settings | **56.8** | 57.3 |
> > | w/ ICL Alignment | 56.2 | **57.9** |
> > | GATEAU-LLaMA - 50% in Limited Short Instruction Data Settings | 59.0 | **54.2** |
> > | w/ ICL Alignment | **59.4** | 53.5 |
> >
> > However, as shown in this table, using the aligned model via in-context learning does not consistently improve the final performance. **This indicates that using only base models in the HMG phase can also achieve good results.** Therefore, HMG can effectively minimize the influence of other factors (e.g., the limited instruction-following ability of a base model) and model the difficulty in modeling the long-range dependencies.
> >
> > > **<4> Is the continual pre-training dataset significantly smaller than the pre-training dataset?**
> >
> > In our method and experiment, the LLaMA-2-7B-base-64k conducts a post-training stage on a total of **10 billion** tokens to extend the context windows. Compared to the pretraining stage of LLaMA-2-7B-base-64k and LLaMA-2-7B-base-4k utilizing about **2 trillion** tokens. **Thus, in our method and experiment, the continual pre-training dataset is significantly smaller than the pre-training dataset and makes long-context LLM $\theta_{B}$ and short-context LLM $\theta_{A}$ have similar other abilities**. Meanwhile, as suggested by previous work[3], continual training on 10B tokens is sufficient for context extension. Thus, in other LLMs, the continual pre-training dataset is always significantly smaller than the pre-training dataset (e.g., LlaMA-3 is pre-trained on a corpus of about 15T tokens).
> >
> > [2] The Unlocking Spell on Base LLMs: Rethinking Alignment via In-Context Learning. ICLR 2024
> >
> > [3] Understanding data influence on context scaling. Yao Fu’s Notion 2023.

---

> ### Author Response · Authors · 2024-11-22
> **Response to Reviewer pvW2 for Weaknesses 2 and Question 1,2,3**
>
> ### **W2:**
> > **W2:** LESS is an optimizer-based method that select a subset of instruction-following dataset by estimating data influence (selecting data points that minimizes the validation loss) and this can be as one of the baselines.
>
> ### **Response:**
>
> LESS is a wonderful and interesting work in SFT data selection. The reason why we do not choose it as the baseline is that LESS needs to extract a subset of the evaluation benchmark to construct a validation set and estimate data influence on this validation set. However, the evaluation benchmarks MT-Bench and LongBench-Chat only contain 80 and 50 test data points respectively. Therefore, extracting a validation set from MT-Bench and LongBench-Chat and using the remaining data as a new test set would severely compromise the final results. This contract the LESS to make a fair comparison with our method and other baselines. We have cited this wonderful work LESS in our related work section.
>
> \\\\
>
> ### **Q1:**
> >  **Q1:** For perplexity guidance (line 377), which $\theta$ do you use ($\theta_{A}$ or $\theta_{B}$ -- I am assuming $\theta_{B}$ here)?
> >
>
> ### **Response:**
> Yes, we use the long-context model  $\theta_{B}$ for baseline Perplexity Guidance. We add a more detailed description in line 377 of our paper (**highlighted in red** in the uploaded revision pdf). Thanks for your advice.
>
> \\\\
>
> ###  **Q2:**
> > **Q2:** What is the context length of MT-Bench? The paper mentions that MT-Bench is for short-context instruction following. Since the proposed method, GATEAU, is designed for long-context, do you have any hypothesis on why it also improves on the short-context instruction following tasks (Table 4)?
> >
>
> ###  **Response:**
> In MT-Bench, the length of the instruction does **not exceed 300 words**. We discuss this interesting  phenomenon in **line 448** of our paper, we conjecture that handling complex tasks (i.e., long input contexts) contributes to handling the easy ones (i.e., short input contexts).
>
> \\\
>
> ### **Q3:**
> > **Q3:** For Table 5, current bold numbers are a bit misleading. Usually bold numbers indicate the highest numbers across some category but here it seems the proposed method is bold. Also, for 13B models, it seem the w/o HMG and w/o CAM settings are not reported. Is there a particular reason of not doing so (e.g., computational constraint)? The ablation study on 7B model does show the effectiveness.
> >
>
> ### **Response:**
> Sorry for the misunderstanding due to our typos in the submission version. We actually want to use bold numbers to indicate the **highest numbers** across methods that use the same ratio of long SFT data. **Now we have modified our paper to correctly show the experimental results**. Meanwhile, as shown in line 477 in our paper, we want to explore whether our method GATEAU can fit in larger LLMs in Table 5. The 13B model (GATEAU-LLaMA-13B) shows consistent improvements on three benchmarks. This indicates that GATEAU scales effectively to larger-scale models. For your concerns, we further conducted the **additional ablation study** for 13B models in our revised paper (**highlighted in red** in the uploaded revision pdf). This indicates the effectiveness of GATEAU and using both two methods can further improve the overall performance as they separately measure the difficulty of generating corresponding responses and understanding long input contexts due to the long-range dependencies.
>
> We sincerely appreciate your careful review and valuable feedback, which have significantly contributed to the improvement of our paper.

---

> ### Author Response · Authors · 2024-11-26
>
> Dear Reviewer pvW2,
>
> We would like to thank you again for your detailed reviews. We have updated our draft and added replies to your concerns with our latest experimental results.
>
> Since the rebuttal deadline is approaching soon. Given that your current score is 5, we would appreciate it if you could let us know if our responses have addressed your concerns satisfactorily. If your concerns have not been resolved, could you please let us know about it so that we have the opportunity to respond before the deadline?
>
> We would be happy to have any follow-up discussions or address any additional concerns.
>
> Looking forward to your reply.
>
> Best,
>
> Authors

---

> ### Comment · Reviewer_pvW2 · 2024-12-02
>
> I thank the authors for providing the response. However, the main concern I have is on the rationale of choosing the base model as a reference model because. I agree that on instruction-following data this can be less of a problem, but in long-context data this can be more of an issue as the base model was not trained on long-context data and the perplexity value is less meaningful. I will keep the current score.

---

> > ### Author Response · Authors · 2024-12-02
> >
> > Dear Reviewer pvW2,
> >
> > Thank you for your valuable feedback. We would like to provide some additional explanations and details to clarify any misunderstandings between us.
> >
> > **<1> Is the perplexity score from the short-context model $\theta_{A}$ really so high that it cannot accurately measure the difficulty?**  We further calculate the average perplexity value generated by the short-context LLM  $\theta_{A}$ for the entire long SFT dataset during the whole HMG process, which is **3.72**. This is because we expand the base frequency of the RoPE position encoding by 200 times (from 10,000 to 2,000,000) to extend the context windows and avoid the model conducting extreme perplexity score (e.g., >1,000) in Homologous Models’ Guidance, detailed in lines 360-366 of our paper.
> >
> >
> >
> > **<2> What Will Happen if We Do not Extend the Context Windows of LLaMA-2-base-4k?**  We further discuss this question in lines 1133-1153 of our paper. As shown in Table 8, we are surprised to find that -w/o Extended Context Windows also achieves competitive results in three benchmarks compared to GATEAU-LLaMA. Even the perplexity score from the short-context model can be very large, e.g., the value can be larger than 1000, the value after softmax normalization is still useful and applicable in the Homologous Models’ Guidance. This interesting finding can be used to reduce the complexity of applying Homologous Models’ Guidance and achieve competitive performance.
> >
> >
> > We hope our response has addressed your concerns, and we look forward to your further feedback.

---

> ### Author Response · Authors · 2024-12-04
>
> Dear Reviewer,
>
> Since it is the last day of the discussion, we hope that you can take a look at our response. Thanks.
>
> Best regards,
>
> The Authors

---

### Official Review · Reviewer_rtwW · 2024-11-11

**Soundness:** 2
**Presentation:** 2
**Contribution:** 3
**Rating:** 5
**Confidence:** 3

**Summary:**

To effectively handle instructions with extremely long contexts in expansion of large language models (LLMs), this paper proposes GATEAU, a novel framework designed to identify the influential samples enriched with long-range dependency relations by utilizing crafted Homologous Models’ Guidance (HMG) and Contextual Awareness Measurement (CAM). Specifically, HMG uses the perplexity scores of the response from two homologous models to measure the ability in modeling long-range dependencies. CAM is used to measure difficulty of understanding the long input contexts by evaluating whether the model’s attention is focused on important segments. Extensive experiments on several LLM benchmarks demonstrate the effectiveness of the proposed algorithms.

**Strengths:**

(1)	This paper proposes an efficient and practical influential-sample-selecting algorithm for long context alignment of large language model. The motivation of the HMG and CAM component is clearly explained in Section 3.

(2)	The organization of this paper is clear and easy to follow. The notations used in Section 3 are all well clarified.

(3)	The experiments are truly extensive, validating the effectiveness of the proposed methods. In particular, the SOTA result in Tables 1-4 is a surprise for me that using part (e.g. 10\%) of the whole dataset can achieve better performance than using the whole dataset. Ablation study in Section 4.3 also verifies the effectiveness of the components (i.e. HMG and CAM).

(4)	Overall, the effort in selecting influential samples for long context alignment in LLM truly makes sense, and the proposed algorithm is simple, motivated, and effective.

**Weaknesses:**

(1)	In Eq.(2), the explanations for the new notation $\theta _{A}$ and $\theta _{B$ should follow immediately after their appearance in Eq.(2).

(2)	In Tables 1-4, the captions should be put at the top (instead of the bottom) of the table.

(3)	In section 4.2 Impact of GATEAU, the paper first analyzes the experimental results in Tables 2 and 4 (line 413), and then analyzed the results in Tables 1 and 3 (line 429). It will be better to analyze the experimental results according to the order of Tables.

(4)	The memory storage and running time of the proposed algorithm is missing, which in my opinion can help readers understand the proposed algorithms more comprehensively. For example, in Eq.(2), HMP model uses short context model $\theta _{A}$ and long context model $\theta _{B}$ to compute the perplexity distance, so does the proposed algorithm have more model parameters (e.g. of  $\theta _{A}$ and $\theta _{B}$ ) than the existing algorithm (e.g. only has $\theta _{B}$)? Therefore, I would suggest to add a table to compare the memory burden as well as the execution time with the existing methods on long context alignment.

(5)	It is a surprise for me that in Table 1 and 3, the proposed algorithm with less dataset can achieve SOTA results in all settings. So, can you give some example (whether simulation or real-world dataset, or theoretical analysis) to explain in what kind of situation the proposed algorithm may fail to achieve SOTA results?

(6)	If I understand correctly, in the proposed algorithm, we need to choose some part of the whole dataset (e.g. 10\% or 30\%) to train the model. How does this paper choose this percentage? And in real-life applications, how do we choose the part (whether 10\% or 30\%) from the whole dataset to align LLM?

**Questions:**

(1)	Could you give some comparisons of memory burden between the proposed algorithms and other exiting methods?

(2)	Could you give some examples to show the failure of the proposed algorithm? It is a little unbelievable that one algorithm can achieve the best performance on all datasets, especially with less training data than existing ones.

(3)	Could you give some insights into the setting of the utilization percentage of the whole dataset of the proposed algorithm?

---

> ### Author Response · Authors · 2024-11-22
> **Response to Reviewer rtwW (1/N)**
>
> Thanks for your valuable review and suggestions! It is encouraging to see you find our paper truly makes sense, and that the proposed method is simple, motivated, and effective.
>
> We sincerely thank you for your time and constructive comments. Below, we provide detailed replies to your comments to resolve your concerns.
>
> ### **W1, W2 and W3:**
> >
> >
> > **W1:** In Eq.(2), the explanations for the new notation \theta _{A} and $\theta _{B}$ should follow immediately after their appearance in Eq.(2).
> >
> > **W2:** In Tables 1-4, the captions should be put at the top (instead of the bottom) of the table.
> >
> > **W3:** In section 4.2 Impact of GATEAU, the paper first analyzes the experimental results in Tables 2 and 4 (line 413), and then analyzed the results in Tables 1 and 3 (line 429). It will be better to analyze the experimental results according to the order of Tables.
> >
>
> ### **Response:**
> Thanks again for your detailed suggestions in our presentation. We have modified our paper in the uploaded revision pdf to make the organization clearer (**highlighted in red** in the uploaded revision pdf).
>
> \\\\
>
> ###  **W4 and Q1:**
> >
> >
> > **W4:** The memory storage and running time of the proposed algorithm is missing, which in my opinion can help readers understand the proposed algorithms more comprehensively. For example, in Eq.(2), HMP model uses short context model $\theta _{A}$  and long context model $\theta _{B}$ to compute the perplexity distance, so does the proposed algorithm have more model parameters (e.g. of $\theta _{A}$ and $\theta _{B}$ ) than the existing algorithm (e.g. only has $\theta _{B}$)? Therefore, I would suggest to add a table to compare the memory burden as well as the execution time with the existing methods on long context alignment.
> >
> > **Q1:** Could you give some comparisons of memory burden between the proposed algorithms and other exiting methods?
>
> ### **Response:**
>
> Thanks for your advice, we believe adding such details will help our paper be more comprehensive.
>
> <1> **Our Experimental Device**: Firstly, we want to explain the experimental device. As shown in Appendix A, all experiments are conducted on 8xA800 80G GPUs (the experiments are not limited to this type of GPU).
>
> <2> **GPU** **Execution Time:** Based on the principle of making full use of GPU devices (e.g., using a multi-processing strategy and choosing large batch sizes, etc, we list the execution time in the following table:
>
> | **Stage** | **Execution Time** |
> | --- | --- |
> | Training on the full dataset in the real-world setting | ~176 GPU hours |
> | Selecting long SFT data via HMG | ~64 GPU hours |
> | Selecting long SFT data via CAM | ~48 GPU hours |
> | Selecting long SFT data via Cherry Selection | ~80 GPU hours |
> | Selecting long SFT data via Perplexity Guidance | ~32 GPU hours |
>
> As shown in this table, we can find our method (HMG + CAM) introduces acceptable offline time overhead compared to the supervised fine-tuning stage and improves the overall performance of long-context LLMs. Perplexity Guidance applies a single LLM to compute the score, thus it achieves less execution time but worse performance in our experiments. Meanwhile, another strong baseline Cherry Selection introduces an additional training stage and computes the proposed Instruction-Following Difficulty (IFD) by applying the forward propagation twice on a single long SFT data, thus necessitating more execution time compared to our proposed HMG. Meanwhile, our CAM and HMG can process the data in parallel to further decrease the execution time, e.g., only 8 hours with 16xA800 80G GPUs.  **Overall, compared to other baselines, the experimental results of our proposed GATEAU (consists of HMG and CAM) demonstrate that the additional execution time is worthwhile.**
>
> <3> **GPU** **Memory Burden:** As our method is designed to score the long SFT data, and then select the influential samples used for the SFT stage, thus our method does not introduce additional memory burden during the supervised fine-tuning and inference stage of the long-context model $\theta_{B}$. For your concerns about HMG, we compute perplexity scores generated from two models $\theta_{A}$ and $\theta_{B}$  for a given SFT data in parallel, and use the computed perplexity scores  (cached in JSON files) to get the homologous models’ perplexity score HMP as shown in Eq. (2). **Thus HMG does not introduce additional GPU memory burden, only introducing acceptable additional execution time as shown in Execution Time Table**. The GPU memory requirements of CAM tem from the calculation of the attention scores for lengthy inputs, as well as the perplexity score computation. **This process is equivalent to performing two forward passes over the dataset without updating gradients, thus it does not add an extra GPU memory burden.**

---

> ### Author Response · Authors · 2024-11-22
> **Response to Reviewer rtwW (2 /N)**
>
> ### **W5 and Q2:**
> >
> >
> > **W5:** It is a surprise for me that in Table 1 and 3, the proposed algorithm with less dataset can achieve SOTA results in all settings. So, can you give some example (whether simulation or real-world dataset, or theoretical analysis) to explain in what kind of situation the proposed algorithm may fail to achieve SOTA results?
> >
> > **Q2:** Could you give some examples to show the failure of the proposed algorithm? It is a little unbelievable that one algorithm can achieve the best performance on all datasets, especially with less training data than existing ones.
> >
>
> ### **Response:**
> Thanks for your interest in the performance of our method. Previous work [1] suggests that data quality is more important than data quantity and shows that instruction tuning  1\% selected high-quality SFT data can outperform the method employing the entire dataset. As mentioned in our paper, previous works attempt to scale up the available data volume by synthesizing long instruction-following samples. However, the absence of a clear strategy for ensuring data quality may lead to the inclusion of low-quality samples. Thus, it is predictable that better performance can be achieved by using fewer but high-quality long SFT data.
>
> Regarding your question, one possible limitation of our method is that our method is designed to improve overall performance in instruction-following and long-context understanding tasks. However, it is not suitable for improving performance in a targeted capability **or task, e.g., only improving the performance of mathematical questions. As shown in Table 7 in the Appendix, our proposed method does not consistently improve the performance across all the different capabilities, e.g., our method achieves unsatisfactory performance in the role-playing task.
>
> [1] One-Shot Learning as Instruction Data Prospector for Large Language Models. ACL 2024
>
> \\\\
>
> ###  **W6 and Q3:**
> >
> >
> > **W6:** If I understand correctly, in the proposed algorithm, we need to choose some part of the whole dataset (e.g. 10% or 30%) to train the model. How does this paper choose this percentage? And in real-life applications, how do we choose the part (whether 10% or 30%) from the whole dataset to align LLM?
> >
> > **Q3:** Could you give some insights into the setting of the utilization percentage of the whole dataset of the proposed algorithm?
> >
>
> ### **Response:**
> Thanks for your interesting and practical question. The ratio of selected long SFT samples is an important hyperparameter in our method. According to the results of experiments, we find our method is robust in the different ratios of used long SFT samples, including 10%, 30%, and 50%. Regarding your question, based on our comprehensive experiments in real-world settings, it is advisable to select 30% of the total data for real-life applications. When computational resources are abundant, we also recommend selecting the optimal ratio by evaluating it in your own real-life scenarios.
>
> \\\\
>
> We hope this detailed explanation clarifies your concerns and underscores the significance of our contributions. If you have any further questions or require additional clarification, please feel free to reach out.

---

> ### Author Response · Authors · 2024-12-04
>
> Dear Reviewer,
>
> Since it is the last day of the discussion, we hope that you can take a look at our response. Thanks.
>
> Best regards,
>
> The Authors

---

### Author Response · Authors · 2024-11-23
**General Response to All Reviewers**

Dear Reviewers,

We thank all reviewers for their insightful comments and acknowledgment of our contributions.

**We greatly appreciate your recognition of the strengths of our work as follows:**

- **Introduction of GATEAU**
    - We present the GATEAU, **as a novel framework(`rtwW`)**, recognized by **all reviewers as well-motivated.**
    - Our framework offers an **efficient and practical** algorithm (**`rtwW`**) to address an **important and essential** challenge(**`KGEd`**,**`93rU`**) for long-context alignment.
- **Methodological Effectiveness**
    - Our method has been acknowledged by **all reviewers** for its **effectiveness, with a clear motivation(`rtwW`,`KGEd`),** which naturally aligns with the challenges of modeling the long-range dependency(**`93rU`**).
- **Comprehensive Experiments**
    - Our Experiments are acknowledged for their **comprehensiveness** by **all reviewers**, validating the **versatility** and **effectiveness** of the proposed methods(**`KGEd`**, **`93rU`**, **`rtwW`**), and showcasing a **well-motivated and interesting**(**`KGEd`**) method for this field.

**We've revised our manuscript per the reviewers' suggestions** (**highlighted in red** in the uploaded revision pdf). Detailed responses to each reviewer's concerns are carefully addressed **point-by-point**.

Below summarize the **major updates** we've made:

- **Presentation:**  We fix the typos in the uploaded revision pdf to make the organization of this paper clearer according to the suggestions from reviewers **`rtwW`** and **`pvW2`**.
- **Experiment:** We further conduct the following experiments to make our paper more sound and try to address reviewers' concerns.
    - **Additional ablation study on 13B models**: (**`pvW2`**). We further conducted the **additional ablation study for 13B models** in our revised paper. This indicates the effectiveness of our proposed GATEAU.
    - **Comparison of execution time**: (**`rtwW`,** **`pvW2`**, and **`93rU`**). We further **compare** the **execution time** of our method with other baselines. The experiment shows that our method introduces **acceptable offline time** overhead compared to the **other baselines** and the SFT stage. Meanwhile, the performance results of our proposed GATEAU (consists of HMG and CAM) demonstrate that the additional execution time is worthwhile.
    - **Exploration of the diversity of selected samples**: (**`93rU`**). We explore the **diversity** of the samples selected by GATEAU. Results show that our proposed GATEAU has partially **addressed sample redundancy** and implicitly **ensured the diversity** of selected long SFT data.
    - **Further exploration of HMG**:(**`pvW2`**) We further explore the designed HMG. As shown in experiments in our response to the **reviewer `pvW2`**, we find that using only base models in the HMG phase can achieve good results. It shows that **HMG can effectively minimize the influence of other factors** (e.g., the limited instruction-following ability of a base model) and model the difficulty in modeling the long-range dependencies.
    - **Further exploration of CAM**: We explore the effect of the number of segments in CAM based on the review from **`KGEd`**. As shown in our experiment **in our response to the reviewer `KGEd`**, different segment lengths affect the model's performance; however, as long as a reasonable length value is chosen, the fluctuations in model performance are not significant. Meanwhile, the performance will always be improved over using the whole long SFT dataset and only using the HMG method, showing the effectiveness of our proposed CAM.
- **Explanation:** We attempt to provide the following explanation to address the misunderstandings in our paper.
    - **GPU memory burden:** We discuss the process of our proposed GATEAU in detail, and explain our proposed GATEAU does **not add an extra GPU memory** burden.
    - **Need for using a single model's perplexity as the sole criterion for data selection:** For the concern from reviewer **`93rU`**, we have already compared this important baseline in the submission version of our paper, namely **Perplexity Guidance**, in our comprehensive experiments (e.g., **Table 1, Table 2, Table 3, Table 4, and Table 8 in the Appendix**) to show our valuable insight, i.e., the high perplexity alone does not adequately reflect response difficulty in long-context scenarios.

We believe our work could make a novel contribution to the community and offer a novel perspective on addressing the challenges of long context alignment of LLM.

We would like to be involved in further discussions if any question is raised.

Best,

Authors.

---

### Comment · Area_Chair_RUAE · 2024-11-25
**Please participate in the discussion with the authors**

Dear Reviewers,

Thank you for your efforts and contribution to ICLR! The authors have posted their responses to your original comments. Only less than two days are left for the reviewer-author discussion. Given the current borderline ratings, your help and prompt responses are important. Please actively check the authors' responses and participate in the discussion.

Thanks!

Best regards,
Your AC

---

### Author Response · Authors · 2024-12-04
**Request for Reviewers to Participate in the Discussion**

Dear Reviewers,

As today is the last day of the discussion, we would like you to take a look at our responses and kindly reconsider the ratings. Thanks.


Best regards,

The Authors

---

### Note · Authors · 2024-12-11

**Comment:**

Dear Reviewers:

We would like to withdraw our submission titled "Selecting Influential Samples for Long Context Alignment via Homologous Models’ Guidance and Contextual Awareness Measurement" (Paper ID: 935) from the ICLR review process. After careful consideration, we have decided to improve the content of our work according to the commnets before resubmitting it to a future venue.

We feel somewhat frustrated that we did not have a full discussion with the reviewers at the rebutaal stage :( . Still, we deeply appreciate the reviewers' time and constructive feedback, which have provided valuable insights for refining our research. Thank you for understanding.

Yours,
Authors

**Withdrawal Confirmation:**

I have read and agree with the venue's withdrawal policy on behalf of myself and my co-authors.